# Unveiling the Hidden Structure: Tight Bounds for Matrix Multiplication Approximation via Convex Optimization

## Abstract

Matrix multiplication lies at the heart of machine learning, yet standard approaches to approximate the multiplication often ignore the interactions that truly governs error. In this work, we introduce a structure-aware upper bound on the optimal achievable approximation using only linear combination of $k$ column-multiplication of the matrices. Our bounds, formulated via convex optimization over an interaction matrix, reveal the hidden challenges and opportunities in matrix multiplication. Through comprehensive numerical experiments, we demonstrate that our bounds not only outperform existing alternatives but also shed new light on the inherent complexity of structured matrix products. This framework paves the way for the development of structure-exploiting algorithms and principled performance guarantees in large-scale machine learning.

**Keywords:** Matrix Multiplication Approximation, Sparse Approximation, Randomized Algorithms, Upper Bounds, Sparsity Constrained Quadratic Programming, Gram Matrix Methods, Convex Optimization

## 1 Introduction

Large-scale matrix multiplication is a computational primitive powering numerous modern machine learning models, including kernel methods (Schölkopf & Smola, 2002; Hofmann et al., 2008), recommender systems (Koren et al., 2009), Transformers (Goodfellow et al., 2016), and graph embeddings. The prohibitive cost of exact multiplication (Strassen, 1969; Coppersmith & Winograd, 1987; Alman & Williams, 2021) motivates Approximate Matrix Multiplication (AMM) (Halko et al., 2011; Mahoney, 2011; Woodruff, 2014). Established AMM techniques, primarily based on sampling (Drineas et al., 2006a) or sketching (Sarlos, 2006; Clarkson & Woodruff, 2017), offer significant computational savings. However, their standard performance guarantees typically depend on coarse matrix properties (e.g., product of Frobenius nomrs of the matrices (Drineas & Mahoney, 2005), stable ranks (Cohen et al., 2016)) and often overlook the fine-grained interaction structure inherent in the product sum.

This oversight becomes critical in many practical ML scenarios where the interplay (cancellations or reinforcements) between the individual vector outer products significantly dictates the true approximation difficulty. This effect is captured by the structure ratio (Section 2). When this ratio is large, indicating strong interactions, standard AMM bounds can become dramatically loose or even *uninformative*, offering little guidance on achievable performance or optimal term selection. As ML models increasingly leverage complex dependencies where such effects are pronounced, we face a fundamental gap: a lack of understanding of the *optimal* error achievable when approximating the product with a budget of only $k$ column-multiplications, especially in these structurally challenging regimes.

Our goal in this work is to provide a computable, instance-specific upper bound on the *optimal* $k$-term Frobenius error. This is distinct from analyzing any specific algorithm, and it yields a structure-aware benchmark that is useful for evaluating *any* $k$-term method on a given instance. For examples of standard AMM bounds (sampling and sketching) and how they differ from our approach, see Section 3 and Table 2. We focus on the Frobenius norm because, crucially, it admits the exact

quadratic representation in Lemma 2.1, which underpins our framework; a comparable formulation for spectral norm is not used here and lies outside our scope.

Our main contributions are:

- Introducing our proposed structure-aware upper bound on the optimal $k$-term approximation error, derived via auxiliary quadratic programs on the interaction matrix.

- Deriving analytical variants of the bound that serve as computationally efficient yet tight approximations to the bound derived from quadratic programs, offering structural insights explicitly dependent on the structure ratio and demonstrating strong empirical performance.

- Demonstrating theoretically and numerically how our proposed bound (and its close approximations) capture approximation difficulty arising from term interactions (quantified by the structure ratio), unlike standard bounds.

The paper is organized as follows: Section 2 formally defines the $k$-term matrix approximation problem and key concepts. Section 3 reviews prior Approximate Matrix Multiplication (AMM) work, highlighting limitations relevant to our objective. Section 4 presents the core derivation of our proposed upper bound, on the optimal $k$-term error. Section 5 provides numerical validation of our bound's tightness and compares it against existing methods.

## 2 PROBLEM FORMULATION

### 2.1 SPARSE APPROXIMATION VIA QUADRATIC OPTIMIZATION

We consider approximating the product $AB^\top = \sum_{j=1}^n a_j b_j^\top$, where $A = (a_1 | \dots | a_n) \in \mathbb{R}^{m \times n}$ and $B = (b_1 | \dots | b_n) \in \mathbb{R}^{p \times n}$. Our goal is to find the best approximation using only $k$ terms, with contributions modulated by a weight vector $x = (x_1, \dots, x_n)^\top \in \mathbb{R}^n$. The approximation is $\widetilde{C}_x = \sum_{j=1}^n x_j a_j b_j^\top$.

The underlying structure governing this approximation is captured by the $n \times n$ *interaction matrix* $G \equiv G_{a,b}$, defined as the Hadamard (element-wise) product of the Gram matrices $G_a = A^\top A$ and $G_b = B^\top B$:

$$G = G_{a,b} = G_a \circ G_b = (A^\top A) \circ (B^\top B). \tag{1}$$

Its entries $[G]_{ij} = \langle a_i, a_j \rangle \langle b_i, b_j \rangle$ quantify the interaction between terms $i$ and $j$. $G$ is symmetric and positive semidefinite (PSD), being the Gram matrix for $\{a_j b_j^\top\}_{j=1}^n$ under the Frobenius inner product. The connection between $G$ and the approximation error is fundamental:

**Lemma 2.1.** *For any weight vector $x \in \mathbb{R}^n$, the squared Frobenius norm error is:*

$$\left\| AB^\top - \sum_{j=1}^n x_j a_j b_j^\top \right\|_F^2 = (\mathbf{1} - x)^\top G (\mathbf{1} - x), \tag{2}$$

*where $\mathbf{1} = (1, \dots, 1)^\top \in \mathbb{R}^n$. The total energy is $\mathbf{1}^\top G \mathbf{1} = \|AB^\top\|_F^2$.*

*Proof.* See Appendix A.1. $\qquad\square$

Lemma 2.1 transforms the AMM problem into minimizing a quadratic form governed by $G$. The core challenge is finding a sparse weight vector $x$ (i.e., $\|x\|_0 \leq k$) within a specified constraint set $K \subseteq \mathbb{R}^n$ (e.g., $K = \{0,1\}^n$ for subset selection, $K = \mathbb{R}_+^n$, $K = \mathbb{R}^n$) that minimizes this error. This defines the generally NP-hard Sparsity Constrained Quadratic Program (SCQP):

$$\mathcal{P}_k(K, G) \quad \begin{cases} \text{minimize} & f(x) = (\mathbf{1} - x)^\top G (\mathbf{1} - x) \\ \text{subject to} & x \in K, \quad \|x\|_0 \leq k \end{cases} \tag{SCQP}$$

We denote the optimal value by $v_k^*(K, G)$. This value represents the *fundamental limit* of $k$-term approximation under constraints $K$. Our central goal is to derive a computable upper bound $u_k^*(K, G)$ for $v_k^*(K, G)$.

## 2.2 THE STRUCTURE RATIO $\rho_G$

The relationship between individual term energies ($\|a_j b_j^\top\|_F^2 = G_{jj}$) and the total energy ($\|AB^\top\|_F^2 = \mathbf{1}^\top G \mathbf{1}$) is crucial. We quantify this using the structure ratio $\rho_G$.

**Definition 2.2.** *Let $G = G_{a,b}$ be the interaction matrix equation 1. Assuming $\|AB^\top\|_F \neq 0$:*

$$\rho_G = \frac{\text{Tr}(G)}{\mathbf{1}^\top G \mathbf{1}} = \frac{\sum_{j=1}^n G_{jj}}{\sum_{i,j} G_{ij}} = \frac{\sum_{j=1}^n \|a_j\|_2^2 \|b_j\|_2^2}{\|AB^\top\|_F^2}. \tag{3}$$

Since $G$ is PSD, $\rho_G \geq 1/n$. Its magnitude reveals the nature of the sum $\sum a_j b_j^\top$. Formally, $\rho_G$ compares the sum of individual term energies (numerator; diagonal of $G$) to the energy of the final product (denominator; full sum over $G$). Thus:

- **Significant cancellation ($\rho_G \gg 1$):** Many anti-aligned pairs yield large negative off-diagonals, making $\mathbf{1}^\top G \mathbf{1} \ll \text{Tr}(G)$ (e.g., $a_i b_i^\top \approx -a_j b_j^\top$); reproducing delicate cancellations makes sparse approximation hard.
- **Strong reinforcement ($\rho_G \approx 1/n$):** Terms are strongly aligned with large positive off-diagonals, so $\mathbf{1}^\top G \mathbf{1} \gg \text{Tr}(G)$; the lower limit $1/n$ occurs when all $n$ term-products are (near) identical.

**Real-World Implications of a structure aware approach.** In practice, a $G$-aware selection is especially beneficial in three common settings: (i) matrices with correlated columns, where near-duplicates ($a_i \approx a_j$) produce large positive $G_{ij}$ and negatives ($a_i \approx -a_j$) produce large negative $G_{ij}$—the former suggests choosing one representative with an adapted weight, while the latter warns against selecting canceling pairs; (ii) structured transforms (e.g., Fourier, wavelets), where many columns have similar norms and norm-based sampling cannot discriminate, but $G$ reveals regular sparse/banded interaction patterns from orthogonality/overlap, enabling a subset that better preserves the signal's essential structure; and (iii) graph-derived matrices with community structure, where $G$ is block-like—norm-based methods overemphasize the largest community, while leveraging $G$ highlights strong intra-community correlations across all blocks and supports balanced, globally informative choices that better preserve graph topology.

| Matrix $M$ | $\text{Tr}(M)$ | $\mathbf{1}^\top M \mathbf{1}$ | Range of $\rho_M$ |
|---|---|---|---|
| $G_a = A^\top A$ | $\|A\|_F^2 = \sum \|a_j\|^2$ | $\|A\mathbf{1}\|^2$ | $[1/n, +\infty)$ |
| $G_{a,b} = G_a \circ G_b$ | $\sum \|a_j\|^2 \|b_j\|^2$ | $\|AB^\top\|_F^2$ | $[1/n, +\infty)$ |
| $G_{a,a} = G_a \circ G_a$ | $\sum \|a_j\|^4$ | $\|AA^\top\|_F^2$ | $[1/n, 1]$ |

Table 1: Trace, Total Sum, and Range of $\rho_M$ for Interaction Matrices.

## 3 RELATED WORK

Existing AMM approaches provide guarantees on specific algorithms but offer limited insight into the fundamental optimal error $v_k^*(K, G)$ of SCQP, especially when interactions are strong ($\rho_G \gg 1$). We review key paradigms, highlighting why their bounds are insufficient as benchmarks for $v_k^*$.

**Sampling-Based AMM.** These methods sample columns $j$ with probabilities $\{p_j\}$ to form $\widetilde{C} = \sum_{l=1}^k \frac{1}{kp_{j_l}} a_{j_l} b_{j_l}^\top$ (Frieze et al., 2004; Achlioptas & Mcsherry, 2007). Optimal probabilities $p_j^{\text{opt}} \propto G_{jj}$ minimize expected error (Drineas et al., 2006a; Drineas & Kannan, 2001), but the resulting bounds depend on $\text{Tr}(G)$ or related sums (e.g., $(\sum \|a_j\| \|b_j\|)^2$), ignoring off-diagonal interactions in $G$. They become uninformative when $\rho_G \gg 1$.

Leverage score sampling (Drineas et al., 2006b; Mahoney & Drineas, 2009) offers powerful alternatives, particularly for regression or spectral norm AMM (Cohen et al., 2016), but these bounds target different objectives and do not directly benchmark the optimal $k$-term Frobenius error $v_k^*(K, G)$

as a function of $G_{a,b}$. Notationally, $\{j_\ell\}_{\ell=1}^k$ denotes $k$ indices sampled *with replacement* from $\{1, \ldots, n\}$ according to $\{p_j\}$; the reweighting by $1/(kp_{j_\ell})$ makes $\widetilde{C}$ an unbiased estimator of $AB^\top$. The standard bounds quoted are on $\mathbb{E}\left[\|AB^\top - \widetilde{C}\|_F^2\right]$.

**Sketching-Based AMM.** Methods using sketching matrices $S \in \mathbb{R}^{k \times n}$ compute $\widetilde{C} = (AS^\top)(SB^\top)$ (Sarlos, 2006; Woodruff, 2014). Popular choices include random projections, Fast JL, SRHT, CountSketch, etc. (Ailon & Chazelle, 2009; TROPP, 2011; Clarkson & Woodruff, 2017; Kane & Nelson, 2014). While efficient, standard Frobenius error bounds typically scale with aggregate norms like $\|A\|_F^2\|B\|_F^2$ (Sarlos, 2006; Cohen et al., 2023). This form inherently ignores the specific product structure $\|AB^\top\|_F^2 = \mathbf{1}^\top G\mathbf{1}$ and the cancellation effects measured by $\rho_G$, limiting their utility as benchmarks for $v_k^*$.

Table 2 lists representative upper bounds for the $k$-term approximation error of $M = AB^\top$. These often rely on $\|M\|_F^2 = \mathbf{1}^\top G\mathbf{1}$, computable in $O(mpn)$ time from the input matrices $A \in \mathbb{R}^{m \times n}$ and $B \in \mathbb{R}^{p \times n}$. For a more comprehensive list and discussion of bounds, see Table A.8.1.

| Method | Frobenius Error Bound | Reference | Key Terms To Compute |
|---|---|---|---|
| Uniform Sampling | $\frac{n}{k}\operatorname{Tr}(G) - \frac{1}{k}\|M\|_F^2$ | (Drineas et al., 2006a) | $\sum G_{ii}, \|M\|_F^2$ |
| Optimal Sampling | $\frac{1}{k}\left(\sum_{i=1}^n \sqrt{G_{ii}}\right)^2 - \frac{1}{k}\|M\|_F^2$ | (Drineas et al., 2006a) | $\sum \sqrt{G_{ii}}, \|M\|_F^2$ |
| Rand. Sketching | $\frac{1}{k}\|A\|_F^2\|B\|_F^2$ | (Sarlos, 2006; Clarkson & Woodruff, 2017) | $\|A\|_F^2, \|B\|_F^2$ |
| Aux QP Bound | $\min_{0 \le s \le k}\left(\|M\|_F^2 + \frac{s}{n}w_s^*(K)\right)$ | Thm. 4.1 | $\|M\|_F^2$ + QP Sol. ($O(n^3)$) |
| Scaled-Id Bound | $\left(1 - \frac{k}{n}\gamma_k^*\right)\|M\|_F^2$ | Prop. 4.3 | $\sum G_{ii}, \|M\|_F^2$ |

Table 2: Representative upper bounds for error $\|AB^\top - \widetilde{C}\|_F^2$ of $k$-term approximate matrix multiplication. For randomized methods the expressions are in expectation.

**Other Guarantees and Methods.** Spectral norm bounds ($\|AB^\top - \widetilde{C}\|$) are crucial for certain applications (Gower & Richtárik, 2015) and leverage tools like matrix concentration and stable rank (Tropp, 2012; Cohen et al., 2016). However, they target a different error measure. Gram matrix approximation ($AA^\top$) methods like Nyström (Williams & Seeger, 2000; Gittens & Mahoney, 2013) or Frequent Directions (Ghashami et al., 2016) focus on low-rank approximation of $AA^\top$, not the general $k$-term selection problem for $AB^\top$. Greedy methods like OMP variants (Tropp & Gilbert, 2007; Needell & Tropp, 2009) or norm-based selection (Belabbas & Wolfe, 2008) are empirically useful but lack guarantees regarding proximity to $v_k^*(K, G)$, and their performance varies significantly with structure ($\rho_G$), precisely where a tight benchmark is most needed.

**Positioning Our Work.** Existing theoretical guarantees predominantly focus on specific algorithms and depend on aggregate properties ($\|A\|_F, \|B\|_F, \operatorname{Tr}(G)$, stable ranks), often failing to reflect the true difficulty imposed by term interactions ($\rho_G$). Our work diverges by directly targeting the *optimal* $k$-term approximation value $v_k^*(K, G)$ itself. By deriving the computable upper bound $u_k^*(K, G)$ from the interaction matrix $G_{a,b}$ (Theorem 4.1), we obtain a *structure-aware* benchmark that depends on the pairwise interactions in $G$, unlike standard bounds (Table 2) based on coarse norms ($\|A\|_F, \|B\|_F$) and thus blind to cancellations measured by $\rho_G$. Because $u_k^*$ is an instance-specific estimate of the best $k$-term error, any heuristic (e.g., OMP, leverage-score sampling) can be evaluated by comparing its empirical error to $u_k^*$ to quantify its gap to optimal. Our experiments (Figures 5–4) demonstrate this, especially in high-cancellation regimes ($\rho_G \gg 1$) where standard bounds are uninformative.

## 4 MAIN RESULTS

We now develop our main contribution: computable upper bounds for the optimal value $v_k^*(K, G)$ of the Sparsity Constrained Quadratic Program SCQP. As highlighted earlier, standard AMM bounds often rely on coarse matrix properties and can become dramatically loose when the interaction structure (quantified by $\rho_G$) leads to significant cancellations or reinforcements. This leaves a critical gap in our understanding: What is the *fundamental limit* of approximation achievable with a $k$-term budget for a *specific* instance $(A, B)$, and how can we quantify it? Our work addresses this by deriving bounds that explicitly incorporate the interaction matrix $G$, providing a much-needed, structure-aware perspective.

The core challenge remains the combinatorial nature of the sparsity constraint $\|x\|_0 \leq k$, which makes problem SCQP NP-hard. Our strategy is to circumvent this direct combinatorial difficulty. We leverage an averaging principle (detailed in the proof appendix) over solutions to related, but *tractable*, continuous optimization problems. These auxiliary problems encode sparsity information and probe the structure of the original objective function.

The central idea is to analyze the objective function landscape $f(x) = (\mathbf{1} - x)^\top G(\mathbf{1} - x)$ by considering approximations that incorporate varying degrees of the interaction structure encoded in the off-diagonal elements of $G$. To achieve this controlled incorporation of structure, we define a sequence of matrices $G^{(s)}$ that interpolate smoothly between the purely diagonal part of $G$ (ignoring all interactions) and the full matrix $G$. Let $\beta_s = \frac{s-1}{n-1}$ for $s = 1, \ldots, n$. We define the matrix:

$$G^{(s)} = \beta_s G + (1 - \beta_s)\text{diag}(G), \quad s = 1, \ldots, n. \tag{4}$$

The matrix $G^{(s)}$ provides a convex interpolation between the diagonal part of $G$ (when $s = 1$) and the full interaction matrix $G$ (when $s = n$). Assuming $G$ is PSD with a positive diagonal (true if no $a_j, b_j$ are zero), $G^{(s)}$ is positive definite for $s < n$. This positive definiteness is crucial as it ensures the strict convexity of the auxiliary quadratic programs equation Aux-QP defined below.

Now, using these interpolated matrices, we define a family of auxiliary quadratic programs. These serve as tractable proxies for the original problem, allowing us to leverage efficient convex optimization techniques. They are parameterized by the interpolation index $s \in \{1, \ldots, n\}$:

$$\mathcal{Q}_s(K) \quad \begin{cases} \text{minimize}_{y \in \mathbb{R}^n} & \phi_s(y) = y^\top G^{(s)} y - 2\langle G\mathbf{1}, y\rangle \\ \text{subject to} & y \in K \end{cases} \tag{Aux-QP}$$

Let $w_s^*(K)$ denote the optimal value of $\mathcal{Q}_s(K)$. We also use the convention $w_0^*(K) = 0$. This problem seeks a vector $y$ within the original constraint set $K$ that minimizes a quadratic objective. The quadratic term $y^\top G^{(s)} y$ is governed by the interpolated interaction matrix $G^{(s)}$, while the linear term $-2\langle G\mathbf{1}, y\rangle$ derives directly from expanding the original objective $f(x)$ around $x = \mathbf{1}$. Because $G^{(s)}$ is typically positive definite or PSD, and the constraint set $K$ is often convex (e.g., $K = \mathbb{R}^n, K = \mathbb{R}_+^n, K = [0, \xi]^n$), the auxiliary problem $\mathcal{Q}_s(K)$ is a convex Quadratic Program (QP). Unlike the original NP-hard SCQP, convex QPs can be solved efficiently using standard optimization techniques (e.g., interior-point methods, active-set methods).

Our main theoretical result establishes the crucial connection: it links the optimal value $v_k^*(K, G)$ of the intractable SCQP problem to the efficiently computable optimal values $w_s^*(K)$ of these tractable auxiliary convex QPs.

### 4.1 UPPER BOUNDS VIA AUXILIARY QPS

**Theorem 4.1** (Upper Bound via Auxiliary QPs). *Assume the constraint set $K \subseteq \mathbb{R}^n$ is convex and has the form $K = \Omega^n$ with $0 \in \Omega$. Let $G$ be a PSD matrix. For any sparsity level $k \in \{1, \ldots, n\}$, the optimal value $v_k^*(K, G)$ of the SCQP problem SCQP is bounded above by:*

$$v_k^*(K, G) \leq u_k^*(K, G) := \min_{0 \leq s \leq k} \left(\mathbf{1}^\top G\mathbf{1} + \frac{s}{n}w_s^*(K)\right). \tag{5}$$

*Proof.* The proof relies on an averaging argument over subsets of indices and upper bound with the expected value of the objective function $\tilde{f}(y_\mathcal{S})$ for subsets $\mathcal{S}$ of cardinality $s$ uniformly chosen in $\{1, \ldots, n\}$, which yields $\mathbf{1}^\top G\mathbf{1} + \frac{s}{n}h(y)$ with $h$ the objective function of of the auxiliary problem $\mathcal{Q}_s(K)$. See Appendix A.3 for the complete derivation. $\square$

**Discussion and Interpretation.** Theorem 4.1 provides a constructive, computationally feasible method to obtain the upper bound $u_k^*(K, G)$ on the optimal $k$-term error $v_k^*(K, G)$. This involves computing interpolated matrices $G^{(s)}$, solving $k$ auxiliary convex QPs $\mathcal{Q}_s(K)$ for their optimal values $w_s^*(K)$, and combining these values via formula equation 5 (details in Algorithm 1, Appendix A.6).

The significance of $u_k^*(K, G)$ lies in its ability to capture instance-specific structure. By incorporating the detailed interactions within $G$ and respecting the weight constraints $K$, it offers a far more informative benchmark than generic, structure-agnostic bounds. It reflects the intrinsic difficulty of achieving a $k$-sparse approximation for the given problem. As experiments in Section 5 will demonstrate, this bound is expected to be tighter than known bounds, and more insightful in high-$\rho_G$ scenarios where simpler bounds falter. Importantly, the minimization over $s \in \{0, \ldots, k\}$ balances two effects: larger $s$ incorporates more of the off-diagonal structure via $G^{(s)}$, but also scales the QP value by $s/n$. Hence the optimizer need not be $s = k$ in general; the trade-off is instance-dependent.

**Analytical Consequences and Role of $\rho_G$.** While the tightest bound $u_k^*(K, G)$ requires solving the auxiliary QPs numerically, we can gain further, more direct insight by analyzing specific feasible solutions for $\mathcal{Q}_s(K)$ or considering specific constraint sets $K$. These analyses lead to *analytical* bounds expressed directly in terms of $n, k$, and the Structure Ratio $\rho_G$. These serve as valuable complements to the numerical bound, offering immediate, interpretable understanding of structural effects.

1. **Binary Case ($K = \{0, 1\}^n$):** Although $K = \{0, 1\}^n$ is not convex, the underlying averaging technique can be adapted for this specific discrete set, yielding a bound directly involving $\rho_G$. (See Appendix A.4 for the adaptation).

**Proposition 4.2** (Binary Case Bound). *For the binary constraint set $K = \{0, 1\}^n$, the optimal value $v_k^*(\{0, 1\}^n, G)$ of SCQP (corresponding to simple subset selection) is bounded by:*

$$v_k^*(\{0, 1\}^n, G) \leq \min \left\{ 1, \left(1 - \frac{k}{n}\right) \left( \left(1 - \frac{k}{n-1}\right) + \frac{k}{n-1} \rho_G \right) \right\} \mathbf{1}^\top G \mathbf{1}. \tag{6}$$

*where $\rho_G = \mathrm{Tr}(G)/(\mathbf{1}^\top G \mathbf{1})$ as defined in equation 3.*

This analytical bound explicitly reveals the influence of the Structure Ratio $\rho_G$.

- If $\rho_G \leq 1$ (low cancellation or reinforcement, e.g., when $G = G_{a,a}$ from Table 2.2), the bound simplifies to $v_k^* \leq (1 - k/n)(1 - \frac{k}{n-1}(1 - \rho_G))\mathbf{1}^\top G \mathbf{1} \leq (1 - k/n)\mathbf{1}^\top G \mathbf{1}$. This guarantees a relative error reduction that is at least linear in $k/n$. This rate is known to be achievable and sharp in simple cases like orthogonal terms ($G \propto I$, where $\rho_G = 1$).

- If $\rho_G > 1$ (significant cancellation). This term acts as a penalty, slowing down the guaranteed rate of error reduction compared to the low-$\rho_G$ case. The bound quantifies how much the difficulty (cancellation) inherent in the problem, measured by $\rho_G$, limits the guaranteed performance of any $k$-subset selection.

The minimum over $s$ allows the bound to adapt; sometimes a smaller $s$ (less interaction considered in $G^{(s)}$) might yield a tighter bound, especially if $k$ is large relative to $n$.

2. **Scaled Identity Analysis ($y = \gamma \mathbf{1}$):** For constraint sets $K$ that contain scaled versions of the all-ones vector $\mathbf{1}$ (e.g., $K = \mathbb{R}^n, K = \mathbb{R}_+^n$), we can analyze the objective $\phi_s(y)$ for the specific feasible point $y = \gamma \mathbf{1}$. Optimizing $\gamma$ for this simple choice within $\mathcal{Q}_s(K)$ provides a suboptimal value for $w_s^*(K)$, which, when plugged into Theorem 4.1, yields another analytical bound. (See Appendix A.5 for derivation).

**Proposition 4.3** (Scaled Identity Bound). *Assume the constraint set $K \subseteq \mathbb{R}^n$ contains the line segment $\{\gamma \mathbf{1} \mid \gamma \in [0, \xi]\}$ where $\xi = \max(1, \rho_G^{-1})$. Let $G$ be PSD. Then for any $k \in \{1, \ldots, n\}$:*

$$v_k^*(K, G) \leq \left(1 - \frac{k}{n} \gamma_k^*\right) \mathbf{1}^\top G \mathbf{1}, \tag{7}$$

*where $\gamma_s^* = \frac{\mathbf{1}^\top G \mathbf{1}}{\mathbf{1}^\top G^{(s)} \mathbf{1}} = \frac{1}{\beta_s + (1 - \beta_s)\rho_G}$ with $\beta_s = \frac{s-1}{n-1}$. The minimum in Theorem 4.1 is achieved at $s = k$ for this specific analysis because $(s/n)\gamma_s^*$ is non-decreasing in $s$.*

*Proof.* The proof involves substituting $y = \gamma\mathbf{1}$ into $\phi_s(y)$, optimizing for $\gamma$ subject to $\gamma \in [0, \xi]$, and using the result in Theorem 4.1. The non-decreasing property of $(s/n)\gamma_s^*$ ensures the minimum occurs at the largest $s$ value considered, i.e., $s = k$. See Appendix A.5. $\qquad\square$

Similar to the binary case, the factor $\gamma_k^*$ modulates the guaranteed linear decay rate based on $\rho_G$.

- If $\rho_G \leq 1$, then $\gamma_k^* \geq 1$. The bound becomes $v_k^*(K, G) \leq (1 - (k/n)\gamma_k^*)\mathbf{1}^\top G\mathbf{1} \leq (1 - k/n)\mathbf{1}^\top G\mathbf{1}$, again recovering the baseline linear decay guarantee.

- If $\rho_G > 1$, then $\gamma_k^* < 1$. The guaranteed relative error reduction is $(k/n)\gamma_k^*$, which is less than $k/n$. The bound explicitly shows a slower guaranteed decay rate, $(1 - (k/n)\gamma_k^*)$, reflecting the increased difficulty imposed by cancellations ($\rho_G > 1$).

This bound applies more broadly than the binary one (to convex $K$ containing $\{\gamma\mathbf{1} : \gamma \in [0, \xi]\}$) but is derived from a specific, potentially suboptimal, choice of $y$ in the auxiliary problem, so it is likely looser than the computationally derived bound $u_k^*(K, G)$ from Theorem 4.1.

**Benchmarking AMM Performance.** Our bounds directly translate to guarantees on the minimal achievable squared Frobenius error for AMM by setting $G = G_{a,b}$ (where $\mathbf{1}^\top G\mathbf{1} = \|AB^\top\|_F^2$). For example, Proposition 4.3 implies for constraint sets $K$ such as $\mathbb{R}_+^n$:

$$\min_{\substack{x \in K \\ \|x\|_0 \leq k}} \left\| AB^\top - \sum_{i=1}^n x_i a_i b_i^\top \right\|_F^2 \leq \left(1 - \frac{k}{n}\gamma_k^*(G_{a,b})\right) \|AB^\top\|_F^2.$$

The tighter bound $u_k^*(K, G_{a,b})$ from Theorem 4.1, computed via the auxiliary QPs, provides the most refined, instance-specific benchmark. Any algorithm (e.g., standard AMM, Sketching, Greedy) producing an error significantly exceeding $u_k^*$ demonstrates *provable suboptimality* for that specific problem instance. This structure-aware benchmark is particularly valuable in high-$\rho_G$ regimes where generic bounds often fail or become ineffective. (Implications for optimal weight norms are detailed in Appendix A.7).

## 5 Experimental Validation: Bound Tightness and Performance Insights

The reported execution times were obtained on a machine with an Intel i7 CPU, 31.2 GB of RAM, and a 233.18 GB disk. All algorithms were implemented as single-core versions. For details of all algorithm implementations refer to Appendix B.5.

In Figure 5, we validate our QP-based bounds against the optimal column selection error, $v_k^*$, for matrices with $n = 30$ across different $\rho_G$ values. Our bounds closely track the optimal solution $v_k^*$, while standard literature bounds become dramatically looser by orders of magnitude, particularly when $\rho_G > 1$. The exhaustive search for $v_k^*$ is only feasible at this small scale ($n = 30$), yet provides crucial validation of our framework. We adapt Orthogonal Matching Pursuit (OMP) to our matrix product approximation problem to demonstrate that our bounds are practically achievable. Despite some instability in the high-$\rho_G$ regime due to cancellation effects and the small number of columns, OMP's performance remains remarkably close to the optimal solution. This confirms that algorithms leveraging the full matrix structure can achieve near-optimal performance, validating our theoretical bounds. For details on matrix generation and OMP adaptation, see Appendices B.1 and B.2.

Figure 2 presents our evaluation across diverse matrix settings ($n = 5000, m = 50, p = 30$), including uniform distributions, Gaussian distributions, and matrices with orthogonal rows. The results demonstrate that our computationally efficient Scaled Id Bound (Prop. 4.3) from Proposition 4.3 consistently and tightly tracks both the more complex Auxiliary QP Bound (Thm. 4.1) (Theorem 4.1) and the empirical performance of OMP. This close alignment persists across fundamentally different matrix structures, confirming the robustness of our Scaled Id Bound (Prop. 4.3) as a practical, easy-to-compute proxy for optimal performance. Notably, OMP consistently outperforms all other methods when $\rho_G$ is not close to zero, highlighting the value of exploiting structural information. Additional experiments with other matrix types are detailed in Appendix B.1.

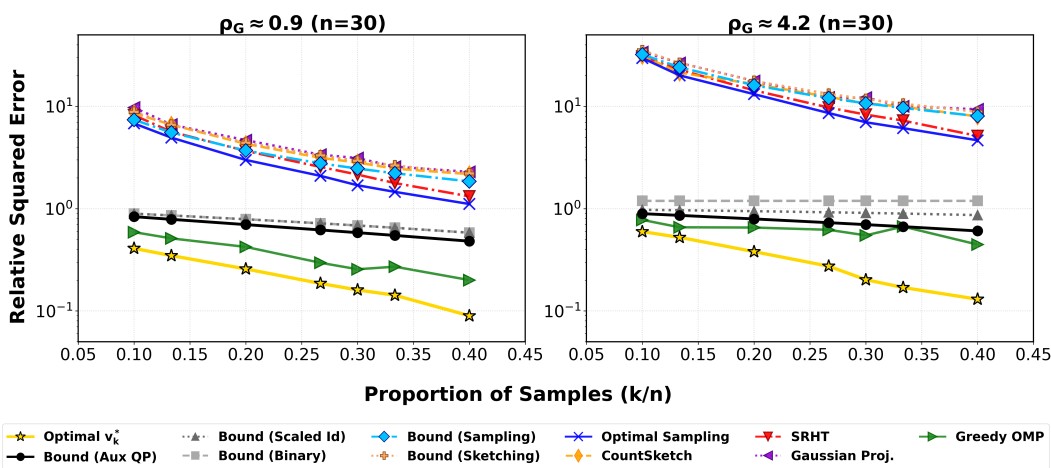

Figure 1: Validation against optimal error ($n = 30$).

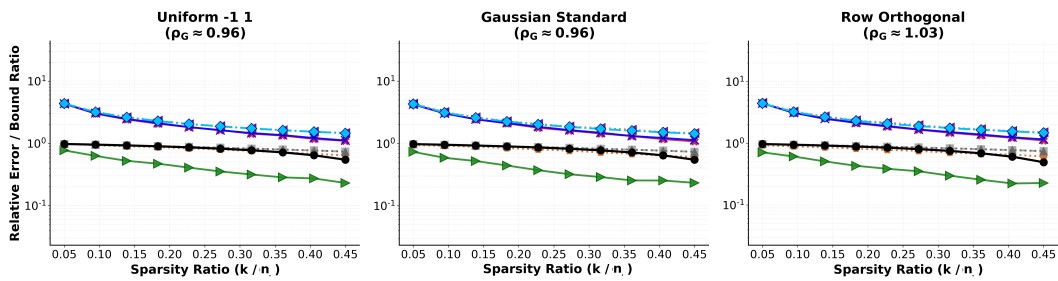

Figure 2: Relative approximation error vs. sampling ratio $k/n$ across different matrix types.

**Scalability Analysis.** Figure 3 ($m = 50, p = 80, k = 0.2n, \rho_G \approx 1$) reveals a critical insight: while existing methods maintain relatively constant error as $n$ increases (suggesting their error depends primarily on the $k/n$ ratio), our bounds and OMP actually improve with larger $n$. The left panel shows OMP achieving the lowest error, closely tracked by our QP-based bounds, while the right panel illustrates the computational trade-off. OMP and the Auxiliary QP Bound (Thm. 4.1) scale more steeply with $n$ than faster sketching or sampling algorithms, reflecting the cost of exploiting structural information. This trade-off is justified when precision is paramount, as our structure-aware approaches deliver substantially better approximations. The efficient Scaled Id Bound (Prop. 4.3) offers a particularly attractive balance, providing much tighter guarantees than generic sketching bounds without the full computational burden of the Auxiliary QP Bound (Thm. 4.1).

Figure 4 (log-log scale) demonstrates how structural complexity $\rho_G$ impacts approximation error across different sampling ratios. As $\rho_G$ increases, standard non-adaptive approaches fail catastrophically, with errors scaling approximately as $\propto \rho_G$. At high $\rho_G$ values, these methods perform no better than uniform random sampling, rendering them essentially uninformative. In stark contrast, OMP maintains consistently low error across all $\rho_G$ values, demonstrating remarkable robustness to structural complexity. Crucially, our QP-based bounds accurately track OMP's empirical performance, confirming their ability to provide reliable performance predictions even in challenging scenarios where traditional bounds become overly pessimistic. This underscores the fundamental importance of structure-aware algorithms and bounds when dealing with complex data interdependencies.

**Practical Implications.** Across all experiments, our Scaled Id Bound (Prop. 4.3) remains remarkably close to the more complex Auxiliary QP Bound (Thm. 4.1) while being significantly more computationally efficient. While computing these bounds requires knowledge of the full product matrix

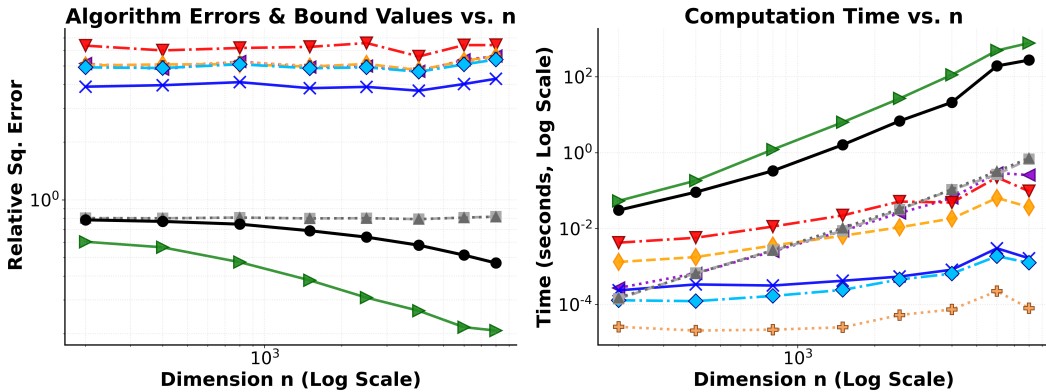

Figure 3: Scalability of approximation algorithms and bounds. Left: Relative error versus matrix dimension $n$. Right: Computation time (seconds, log-log scale) versus $n$. For this experiment, $m = 50, p = 80, k = 0.2n$, and matrices $A, B$ were generated from an i.i.d. Gaussian distribution, resulting in $\rho_G \approx 1$.

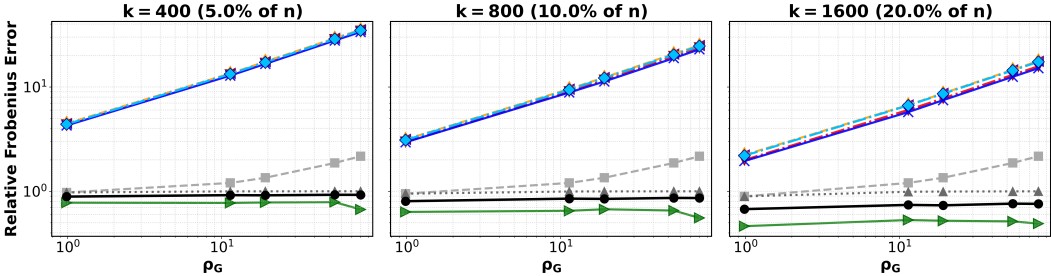

Figure 4: Impact of structural complexity ($\rho_G$) on approximation error (log-log scale) for ranks $k = 0.05n$ (left), $k = 0.1n$ (middle), and $k = 0.2n$ (right) ($n = 8000, m = 30, p = 50$).

(making them primarily valuable as theoretical tools rather than runtime optimizations), they provide crucial insights for algorithm design. Our results strongly suggest that developing new structure-aware algorithms that can exploit the information captured by the matrix $G = (A^T A) \odot (B^T B)$ represents a promising direction for future research. Such algorithms could potentially achieve the accuracy benefits demonstrated by OMP while maintaining better computational scalability, opening new avenues for efficient matrix product approximation in high-dimensional settings.

**Reproducibility Statement.** Reproducibility is supported by: clear problem setup, notation, and assumptions in Section 2 (incl. $G$ and $\rho_G$) and Lemma 2.1; complete proofs for Theorem 4.1 and Propositions 4.2–4.3 in the appendix; implementation details and solver settings for computing the bounds in Appendix A.6; and fully specified experimental protocols, data generation, OMP pseudocode, and hyperparameters in Section 5 and Appendices B.1, B.2, B.5. An anonymized code archive with scripts to generate figures is included in the supplementary materials.

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

LLM Usage Disclosure

LLMs were used only for light copy-editing and not for research ideation, proofs, code, or experiments. The authors take full responsibility for all content; LLMs are not authors.

# A  Appendix: Proofs and Additional Details

This appendix provides detailed proofs for the main theoretical results presented in the paper, along with supplementary discussions, algorithmic details, and analysis of special cases.

## A.1  Proof of Lemma 2.1

We want to show that $\|AB^\top - \sum_{j=1}^n x_j a_j b_j^\top\|_F^2 = (\mathbf{1}-x)^\top G(\mathbf{1}-x)$. Since $AB^\top = \sum_{j=1}^n a_j b_j^\top$, then $E_x := AB^\top - \sum_{j=1}^n x_j a_j b_j^\top = \sum_{j=1}^n (1-x_j) a_j b_j^\top$. Let $y = \mathbf{1} - x$, where $\mathbf{1}$ is the vector of all ones. Then $y_j = 1 - x_j$, hence $E_x = \sum_{j=1}^n y_j a_j b_j^\top$.

The squared Frobenius norm is defined as $\|E_x\|_F^2 = \mathrm{Tr}(E_x^\top E_x)$. Let's compute $E_x^\top E_x$:

$$
E_x^\top E_x = \left( \sum_{i=1}^n y_i a_i b_i^\top \right)^\top \left( \sum_{j=1}^n y_j a_j b_j^\top \right)
$$

$$
= \left( \sum_{i=1}^n y_i b_i a_i^\top \right) \left( \sum_{j=1}^n y_j a_j b_j^\top \right)
$$

$$
= \sum_{i=1}^n \sum_{j=1}^n y_i y_j b_i (a_i^\top a_j) b_j^\top.
$$

Now, we take the trace:

$$
\mathrm{Tr}(E_x^\top E_x) = \mathrm{Tr}\left( \sum_{i=1}^n \sum_{j=1}^n y_i y_j b_i (a_i^\top a_j) b_j^\top \right)
$$

$$
= \sum_{i=1}^n \sum_{j=1}^n y_i y_j (a_i^\top a_j) \, \mathrm{Tr}(b_i b_j^\top) \quad \text{(Linearity of Trace)}
$$

$$
= \sum_{i=1}^n \sum_{j=1}^n y_i y_j (a_i^\top a_j)(b_j^\top b_i) \quad \text{(Using } \mathrm{Tr}(uv^\top) = v^\top u)
$$

$$
= \sum_{i=1}^n \sum_{j=1}^n y_i y_j \langle a_i, a_j \rangle \langle b_i, b_j \rangle
$$

$$
= \sum_{i=1}^n \sum_{j=1}^n y_i y_j [G]_{ij} \quad \text{(By definition of } G = (A^\top A) \odot (B^\top B))
$$

$$
= y^\top G y = (\mathbf{1}-x)^\top G (\mathbf{1}-x).
$$

This proves the first part of the lemma.

For the second part, plugging $x = \mathbf{0}$ into the formula we just derived gives $\|AB^\top\|_F^2 = \mathbf{1}^\top G \mathbf{1}$. This confirms the second statement of the lemma.

## A.2  Properties of the Gram Matrix G

The analysis of our bounds and the Greedy OMP algorithm relies heavily on the properties of the Gram matrix $G = G_{a,b} = (A^T A) \odot (B^T B)$, where $\odot$ denotes the element-wise (Hadamard) product.

The matrix $G$ is symmetric and positive semi-definite (SDP) because it is a Gram matrix associated with the set of matrices $\{a_i b_i^\top\}_{i=1}^n$ when considering the Frobenius inner product as an inner product

in the space of matrices. It is positive definite if and only if the matrices $\{a_i b_i^\top\}_{i=1}^n$ are linearly independent. This condition may not hold if $pm < n$.

**Remark A.1** (Special Cases for Positive Definiteness). *The following cases ensure linear independence and thus positive definiteness of $G$:*

- *If $\mathrm{rank}(A) = n$ and $B$ has no zero columns.*

- *If $\mathrm{rank}(B) = n$ and $A$ has no zero columns.*

*We prove the first case; the second follows by transposition. Let $x \in \mathbb{R}^n$ be such that $\sum_{i=1}^n x_i a_i b_i^\top = 0$. Taking the inner product with $a_k b_k^\top$ yields $\sum_{i=1}^n x_i \langle a_i b_i^\top, a_k b_k^\top \rangle = \sum_{i=1}^n x_i \langle a_i, a_k \rangle \langle b_i, b_k \rangle = 0$ for all $k \in [n]$. This can be written as $(Gx)_k = 0$ for all $k$. Alternatively, multiplying by $b_k$ on the right gives $\sum_{i=1}^n x_i a_i b_i^\top b_k = (\sum_{i=1}^n x_i a_i b_i^\top) b_k = 0$. This implies $\sum_{i=1}^n (x_i b_k^T b_i) a_i = 0$. Since $\mathrm{rank}(A) = n$, the vectors $a_i$ are linearly independent, so we must have $x_i b_k^T b_i = 0$ for all $i$. As $B$ has no zero columns, $b_k^T b_k = \|b_k\|^2 \neq 0$, thus $x_k = 0$. We then repeat for all $k$.*

In general, definiteness of $G$ can still occur even when $A^T A$ and $B^T B$ are singular (e.g., when $m < n$ and $p < n$). The inequality below relating the ranks $r_a = \mathrm{rank}(A)$, $r_b = \mathrm{rank}(B)$ and $r_{a,b} = \mathrm{rank}(G)$ is established in Horn & Yang (2020),

$$r_{a,b} \geq \min(n, \max(r_a + \widehat{r}_b, r_b + \widehat{r}_a) - 1), \tag{8}$$

where $\widehat{r}_a$ and $\widehat{r}_b$ denote the Kruskal ranks of $A$ and $B$, respectively, as defined in Horn & Yang (2020).

The Kruskal rank of an $m \times n$ matrix $X$ that has no zero columns is defined as the largest positive integer $k$ such that every list of $k$ distinct columns of $X$ is linearly independent. This definition corresponds to $\mathrm{spark}(X)$, the spark of $X$ (smallest number of linearly dependent columns). Using the facts $\mathrm{rank}(X^\top X) = \mathrm{rank}(X)$ and that the Kruskal rank relates to spark, we can interpret equation 8. A related inequality might be formulated using spark directly, potentially leading to:

$$\mathrm{rank}(G) \geq \min(n, \max(\mathrm{rank}(A) + \mathrm{spark}(B) - 1, \mathrm{spark}(A) + \mathrm{rank}(B) - 1)). \tag{9}$$

This suggests conditions like $\mathrm{rank}(A) + \mathrm{spark}(B) \geq n + 1$ or $\mathrm{rank}(B) + \mathrm{spark}(A) \geq n + 1$ might contribute to ensuring $G$ is nonsingular.

For the SDP matrix $G$, the trace $\mathrm{Tr}(G)$ (sum of diagonal elements) and the total sum $\mathbf{1}^\top G \mathbf{1}$ (sum of all elements) are crucial. We define the structure ratio $\rho_G$ as

$$\rho_G := \frac{\mathrm{Tr}(G)}{\mathbf{1}^\top G \mathbf{1}} = \frac{\sum_{i=1}^n \|a_i\|^2 \|b_i\|^2}{\|AB^\top\|_F^2}. \tag{10}$$

We assume $\mathbf{1}^\top G \mathbf{1} = \|AB^\top\|_F^2 \neq 0$, otherwise the approximation problem is trivial. Since $\mathbf{1}^\top G \mathbf{1} \leq n \mathrm{Tr}(G)$ (via Cauchy-Schwarz or Gershgorin bounds on eigenvalues), we have $\rho_G \geq 1/n$. The ratio $\rho_G$ can be arbitrarily large, particularly if $G$ is singular and $\mathbf{1}$ has significant components in the null space or near-null space of $G$.

### A.3 PROOF OF THEOREM 4.1 (MAIN UPPER BOUND)

Let $x^*$ be an optimal solution of $\mathcal{P}_k(K, G)$, and let $y \in K$ be an arbitrary fixed vector in the set $K$. For any subset $\mathcal{S} \subset [n]$ of size $s \in \{1, \ldots, k\}$, consider the vector $y_\mathcal{S}$. Since $K = \Omega^n$ with $0 \in \Omega$, the vector $y_\mathcal{S}$ which coincides with $y$ on $\mathcal{S}$ and is zero elsewhere belongs to $K$. That is, $(y_\mathcal{S})_i = y_i$ if $i \in \mathcal{S}$ and $(y_\mathcal{S})_i = 0$ if $i \notin \mathcal{S}$. Since $y_\mathcal{S}$ is feasible (as $\|y_\mathcal{S}\|_0 \leq s \leq k$ and $y_\mathcal{S} \in K$), then

$$v_k^*(K, G) = (\mathbf{1} - x^*)^\top G (\mathbf{1} - x^*) \leq (\mathbf{1} - y_\mathcal{S})^\top G (\mathbf{1} - y_\mathcal{S})$$
$$= \mathbf{1}^\top G \mathbf{1} - 2 \, \mathbf{1}^\top G y_\mathcal{S} + y_\mathcal{S}^\top G y_\mathcal{S}$$
$$= v_0 - 2 \langle q, y_\mathcal{S} \rangle + y_\mathcal{S}^\top G y_\mathcal{S},$$

where $v_0 = \mathbf{1}^\top G \mathbf{1}$ and $q = G \mathbf{1}$. Summing over all $\binom{n}{s}$ possible sets $\mathcal{S}$ of size $s$, we bound $v_k^*(K, G)$ according to

$$\binom{n}{s}(v_k^*(K, G) - v_0) \leq \sum_{\mathcal{S} \subset [n], |\mathcal{S}| = s} \left( -2 \langle q, y_\mathcal{S} \rangle + y_\mathcal{S}^\top G \, y_\mathcal{S} \right).$$

We then expand the sums on the right-hand side. For the linear term, since each index $i$ belongs to $\binom{n-1}{s-1}$ sets $\mathcal{S}$ of size $s$, we have

$$\sum_{\substack{\mathcal{S} \subset [n] \\ |\mathcal{S}| = s}} \langle q, y_\mathcal{S} \rangle = \sum_{\substack{\mathcal{S} \subset [n] \\ |\mathcal{S}| = s}} \sum_{i \in \mathcal{S}} q_i y_i = \sum_{i=1}^{n} \sum_{\substack{\mathcal{S} \subset [n] \\ |\mathcal{S}| = s, i \in \mathcal{S}}} q_i y_i = \binom{n-1}{s-1} \sum_{i=1}^{n} q_i y_i = \binom{n-1}{s-1} \langle q, y \rangle.$$

For the quadratic term $y_\mathcal{S}^\top G y_\mathcal{S} = \sum_{i,j \in \mathcal{S}} G_{ij} y_i y_j = \sum_{i \in \mathcal{S}} G_{ii} y_i^2 + \sum_{i \neq j \in \mathcal{S}} G_{ij} y_i y_j$, we need to consider separately the two sums. When summing the first for all $\mathcal{S}$, each $G_{ii} y_i^2$ is added $\binom{n-1}{s-1}$ times. On the second summation, each $G_{ij} y_i y_j$ ($i \neq j$) is added $\binom{n-2}{s-2}$ times (for $s \geq 2$). Therefore,

$$\sum_{\substack{\mathcal{S} \subset [n] \\ |\mathcal{S}| = s}} y_\mathcal{S}^\top G y_\mathcal{S} = \binom{n-1}{s-1} \sum_{i=1}^{n} G_{ii} y_i^2 + \binom{n-2}{s-2} \sum_{i \neq j \in [n]} G_{ij} y_i y_j.$$

Combining all the previous and using identities

$$\binom{n-1}{s-1} = \frac{s}{n} \binom{n}{s}, \qquad \binom{n-2}{s-2} = \frac{s(s-1)}{n(n-1)} \binom{n}{s},$$

valid for $s = 2, \ldots, n$, we imply

$$v_k^*(K, G) - v_0 \leq \frac{s}{n} \left( -2 \langle q, y \rangle + \sum_{i=1}^{n} G_{ii} y_i^2 + \frac{s-1}{n-1} \sum_{i \neq j \in [n]} G_{ij} y_i y_j \right).$$

We recall the notation $\beta_s = (s-1)/(n-1)$. Since $G^{(s)}$ has the same diagonal as $G$ and its off-diagonal elements are $\beta_s G_{ij}$, the quadratic form above is $\sum_{i=1}^{n} G_{ii} y_i^2 + \beta_s \sum_{i \neq j \in [n]} G_{ij} y_i y_j = y^\top G^{(s)} y$. Thus, the expression in the parenthesis is reduced to $-2 \langle q, y \rangle + y^\top G^{(s)} y$. We note that for $s = 1$, the same arguments apply, yielding $-2 \langle q, y \rangle + \sum_{i=1}^{n} G_{ii} y_i^2 = -2 \langle q, y \rangle + y^\top G^{(1)} y$ since $G^{(1)} = G$. Since the inequality $v_k^*(K, G) - v_0 \leq \frac{s}{n} (-2 \langle q, y \rangle + y^\top G^{(s)} y)$ holds for any $y \in K$, minimizing the right hand side over $y \in K$ implies

$$v_k^*(K, G) \leq v_0 + \frac{s}{n} \min_{y \in K} \left( -2 \langle q, y \rangle + y^\top G^{(s)} y \right) = v_0 + \frac{s}{n} w_s^*(K) = u_s.$$

Minimizing over all $s \in \{1, \ldots, k\}$ yields $v_k^*(K, G) \leq \min_{1 \leq s \leq k}(u_s)$. Including the trivial bound $v_k^*(K, G) \leq v_0 = \mathbf{1}^\top G \mathbf{1}$ (corresponding to $x = 0$), we get $v_k^*(K, G) \leq \min(v_0, \min_{1 \leq s \leq k} u_s)$. The theorem statement uses $u_0 = v_0$ and defines $u_s = v_0 + \frac{s}{n} w_s^*(K)$, so the bound is $\min_{0 \leq s \leq k} u_s$. The proof is complete.

**Trade-off and Approximation** There is an inherent trade-off in this approach. We are replacing a combinatorial optimization problem with a set of continuous optimization problems. The quality of the upper bound depends on how well the solutions of $\mathcal{Q}_s(K)$ approximate the behavior of the original problem $\mathcal{P}_k(K, G)$. In cases where the structure of $G$ and $K$ allows for efficient solution of $\mathcal{Q}_s(K)$, this method provides a practical approximation strategy and a useful theoretical benchmark.

### A.4 Proof of Proposition 4.2 (Binary Case Bound)

For $s \in \{1, \ldots, n\}$ fixed, we consider problem $\mathcal{Q}_s(\{0, 1\}^n)$. Remark that:

$$\mathbf{1}^\top G^{(s)} \mathbf{1} = \beta_s \mathbf{1}^\top G \mathbf{1} + (1 - \beta_s) \text{Tr}(G), \qquad s = 1, \ldots, n. \tag{11}$$

The vector $\mathbf{1}$ is feasible for this problem (since $\mathbf{1} \in \{0, 1\}^n$). Thus, the optimal value $w_s^*(\{0, 1\}^n)$ satisfies $w_s^*(\{0, 1\}^n) \leq \phi_s(\mathbf{1}) = \mathbf{1}^\top G^{(s)} \mathbf{1} - 2 \langle G \mathbf{1}, \mathbf{1} \rangle = \mathbf{1}^\top G^{(s)} \mathbf{1} - 2 \times \mathbf{1}^\top G \mathbf{1}$. Using equation equation 11 for $\mathbf{1}^\top G^{(s)} \mathbf{1}$, we get:

$$w_s^*(\{0, 1\}^n) \leq (\beta_s \mathbf{1}^\top G \mathbf{1} + (1 - \beta_s) \text{Tr}(G)) - 2 \times \mathbf{1}^\top G \mathbf{1}$$
$$= -\mathbf{1}^\top G \mathbf{1} + (1 - \beta_s)(\text{Tr}(G) - \mathbf{1}^\top G \mathbf{1}).$$

Now, we use this in the general bound from Theorem 4.1: We have

$$\mathbf{1}^\top G\mathbf{1} + \frac{s}{n}w_s^*(\{0,1\}^n) \leq u_s := \mathbf{1}^\top G\mathbf{1} + \frac{s}{n}[-\mathbf{1}^\top G\mathbf{1} + (1-\beta_s)(\text{Tr}(G) - \mathbf{1}^\top G\mathbf{1})]$$

$$u_s = \left(1 - \frac{s}{n}\right)\mathbf{1}^\top G\mathbf{1} + \frac{s}{n}(1-\beta_s)(\text{Tr}(G) - \mathbf{1}^\top G\mathbf{1})$$

$$u_s = \left(1 - \frac{s}{n}\right)\mathbf{1}^\top G\mathbf{1} + \frac{s}{n}\left(\frac{n-1-(s-1)}{n-1}\right)(\text{Tr}(G) - \mathbf{1}^\top G\mathbf{1})$$

$$u_s = \left(1 - \frac{s}{n}\right)\mathbf{1}^\top G\mathbf{1} + \frac{s(n-s)}{n(n-1)}(\text{Tr}(G) - \mathbf{1}^\top G\mathbf{1})$$

$$u_s = \left(1 - \frac{s}{n}\right)\mathbf{1}^\top G\mathbf{1} + \frac{s}{n-1}\left(1 - \frac{s}{n}\right)(\text{Tr}(G) - \mathbf{1}^\top G\mathbf{1})$$

$$u_s = \left(1 - \frac{s}{n}\right)\left[\mathbf{1}^\top G\mathbf{1} + \frac{s}{n-1}(\text{Tr}(G) - \mathbf{1}^\top G\mathbf{1})\right]$$

$$u_s = \left(1 - \frac{s}{n}\right)\left[\left(1 - \frac{s}{n-1}\right)\mathbf{1}^\top G\mathbf{1} + \frac{s}{n-1}\text{Tr}(G)\right]$$

$$u_s = \left(1 - \frac{s}{n}\right)\left[\left(1 - \frac{s}{n-1}\right) + \frac{s}{n-1}\rho_G\right]\mathbf{1}^\top G\mathbf{1}.$$

Theorem 4.1 applied for $K = \{0,1\}^n$ implies that $v_k^*(\{0,1\}^n, G)$ is bounded by $\min_{0 \leq s \leq k}(u_s)$. For $n, \rho > 0$ fixed, the function $t \mapsto \left(1 - \frac{t}{n}\right)\left(\frac{t}{n-1}\rho + (1 - \frac{t}{n-1})\right)$ has a simple monotony pattern over $[0, n]$. It is decreasing if $\rho \leq 2 - 1/n$, otherwise it increases then decreases. As a result, we have $\min_{0 \leq s \leq k}(u_s) = \min(u_0, u_k)$. The proof is complete.

**Discussion on Proposition 4.2** In the problem of approximating $AB^\top$, the sharpness of the estimate $\min(u_0, u_k)$ (where $u_s$ is given by the proof above) is observed when $G$ is, for example, the identity matrix, corresponding, for instance, to the case where the rank-one matrices $a_i b_i^\top$ are pairwise orthogonal and normalized ($\|a_i b_i^\top\|_F = 1$). In this specific scenario ($\text{Tr}(G) = n$, $\mathbf{1}^\top G\mathbf{1} = n$, $\rho_G = 1$), the upper bound equation 6 simplifies to $u_k = \left(1 - \frac{k}{n}\right)\mathbf{1}^\top G\mathbf{1}$. This bound is achieved by selecting any $k$ indices, yielding an error of $\sum_{n-k \text{ terms}} G_{jj} = n - k = (1 - k/n)n = (1 - k/n)\mathbf{1}^\top G\mathbf{1}$.

It is worth noting that the inequality $u_k \leq (1 - k/n)\mathbf{1}^\top G\mathbf{1}$ remains valid in more general cases, specifically when $\rho_G \leq 1$ (i.e. $\text{Tr}(G) \leq \mathbf{1}^\top G\mathbf{1}$). This condition is satisfied, for example, if $\langle a_i, a_j \rangle$ and $\langle b_i, b_j \rangle$ have consistent signs (or one is zero) for all pairs $i \neq j$, ensuring $G_{ij} \geq 0$.

We note that

$$\left(1 - \frac{s}{n-1}\right) + \frac{s}{n-1}\frac{1}{n} = 1 - \frac{s}{n-1}\left(1 - \frac{1}{n}\right) = 1 - \frac{s}{n}.$$

Therefore

$$u_s = \left(1 - \frac{s}{n}\right)\left[\left(1 - \frac{s}{n}\right) + \frac{s}{n-1}\left(\rho_G - \frac{1}{n}\right)\right]\mathbf{1}^\top G\mathbf{1}.$$

In extreme cases where $\rho_G = 1/n$ (implying $\mathbf{1}^\top G\mathbf{1} = n\,\text{Tr}(G)$ which occurs if and only if $G$ is proportional to $\mathbf{1}\mathbf{1}^\top$), the upper bound $\min(u_0, u_k)$ further simplifies to $(1 - k/n)^2\,\mathbf{1}^\top G\mathbf{1}$. This yields the bound:

$$v_k^*(\{0,1\}^n, G) \leq \left(1 - \frac{k}{n}\right)^2 \mathbf{1}^\top G\mathbf{1}, \tag{12}$$

This extremal setting corresponds to scenarios where the matrices $a_i b_i^\top$ are highly correlated, specifically $a_i b_i^\top = \lambda_i M$ for some fixed rank-one matrix $M$ and scalars $\lambda_i$. If $\lambda_i > 0$ for all $i$, optimal solutions indeed achieve an objective value bounded by $\left(1 - \frac{k}{n}\right)^2 \mathbf{1}^\top G\mathbf{1}$, with equality when all $\lambda_i$ are equal. This is discussed further in Appendix A.8.

A.5  PROOF OF PROPOSITION 4.3 (SCALED IDENTITY BOUND)

We explore solutions to the quadratic program $\mathcal{Q}_s(K)$ of the form $y = \gamma\mathbf{1}$ with $\gamma \geq 0$. Substituting $y = \gamma\mathbf{1}$ into the objective function of $\mathcal{Q}_s(K)$ equation Aux-QP, we aim to minimize over $\gamma$. The objective function becomes:

$$\phi_s(\gamma\mathbf{1}) = (\gamma\mathbf{1})^\top G^{(s)}(\gamma\mathbf{1}) - 2\langle G\mathbf{1}, \gamma\mathbf{1}\rangle = (\mathbf{1}^\top G^{(s)}\mathbf{1})\gamma^2 - 2(\mathbf{1}^\top G\mathbf{1})\gamma.$$

Since $G^{(s)}$ is positive definite for $s < n$ (assuming $G_{ii} > 0$), we have $\mathbf{1}^\top G^{(s)}\mathbf{1} > 0$. Minimizing the quadratic function with respect to $\gamma \in \mathbb{R}$ yields the optimal solution $\gamma_{unc}^* = \frac{\mathbf{1}^\top G\mathbf{1}}{\mathbf{1}^\top G^{(s)}\mathbf{1}}$. The optimal value associated with this choice is

$$\phi_s(\gamma_{unc}^*\mathbf{1}) = (\mathbf{1}^\top G^{(s)}\mathbf{1})(\gamma_{unc}^*)^2 - 2(\mathbf{1}^\top G\mathbf{1})\gamma_{unc}^* = -(\mathbf{1}^\top G\mathbf{1})\gamma_{unc}^*$$

Let $\gamma_s^* = \gamma_{unc}^*$. Recall the definition $\rho_G = \mathrm{Tr}(G)/\mathbf{1}^\top G\mathbf{1}$ (assuming $\mathbf{1}^\top G\mathbf{1} > 0$). Using equation 11, we express $\gamma_s^*$ as:

$$\gamma_s^* = \frac{\mathbf{1}^\top G\mathbf{1}}{\beta_s(\mathbf{1}^\top G\mathbf{1}) + (1 - \beta_s)\mathrm{Tr}(G)} = \frac{1}{\beta_s + (1 - \beta_s)\rho_G}, \qquad 1 \leq s \leq n. \tag{13}$$

We recall that the ratio $\rho_G = \mathrm{Tr}(G)/\mathbf{1}^\top G\mathbf{1}$ belongs to $[1/n, +\infty[$. We have that $\gamma_1^* = 1/\rho_G \in ]0, n]$ (since $\beta_1 = 0$) and $\gamma_s^* \in ]0, 1/\beta_s]$ for $s \in \{2, \ldots, n\}$, with $\gamma_n^* = 1$ (since $\beta_n = 1$). By simply using $\beta_s + (1 - \beta_s)\frac{1}{n} = \frac{1}{n} + \beta_s\frac{n-1}{n} = \frac{s}{n}$, we rearrange the above expression as

$$\gamma_s^* = \frac{1}{\dfrac{s}{n} + (1 - \beta_s)\left(\rho_G - \dfrac{1}{n}\right)}, \qquad 1 \leq s \leq n.$$

Since $\rho_G \geq 1/n$, we can easily check that the sequence $(\frac{s}{n}\gamma_s^*)_{s=1}^n$ is non-decreasing. It is strictly increasing if $\rho_G > 1/n$ and constant (equal to 1) if $\rho_G = 1/n$.

Combining the analysis above with Theorem 4.1, we derive Proposition 4.3.

*Proof of Proposition 4.3.* The assumption is that $K \subseteq \mathbb{R}^n$ contains the line segment $\{\gamma\mathbf{1} \mid \gamma \in [0, \xi]\}$ where $\xi = \max(1, \rho_G^{-1})$. We need to ensure that the unconstrained minimizer $\gamma_s^*$ lies within the feasible range $[0, \xi]$ for the $\gamma$ variable when considering the restricted problem $\min_{\gamma \in [0, \xi]} \phi_s(\gamma\mathbf{1})$.

Case 1: $\rho_G \leq 1$. Then $\xi = \rho_G^{-1} \geq 1$. In this case $\gamma_s^*$ is decreasing in $s$, hence $\gamma_s^* \in [0, \gamma_1^*] = [0, \rho_G^{-1}]$ for all $1 \leq s \leq n$.

Case 2: $\rho_G > 1$. Then $\xi = 1$. In this case $\gamma_s^* \leq 1$ (since $\beta_s + (1 - \beta_s)\rho_G \geq \beta_s + (1 - \beta_s) = 1$) for all $1 \leq s \leq n$.

In both cases, $\gamma_s^* \in [0, \xi]$. Therefore, the minimum of $\phi_s(\gamma\mathbf{1})$ over $\gamma \in [0, \xi]$ occurs at $\gamma = \gamma_s^*$. This implies that for the original problem $\mathcal{Q}_s(K)$, the optimal value $w_s^*(K)$ must be less than or equal to the value achieved by the feasible point $y = \gamma_s^*\mathbf{1}$.

$$w_s^*(K) \leq \phi_s(\gamma_s^*\mathbf{1}) = -(\mathbf{1}^\top G\mathbf{1})\gamma_s^*$$

for any $s \in \{1, \ldots, n\}$. We then apply Theorem 4.1:

$$v_k^*(K, G) \leq \min_{0 \leq s \leq k} \left(\mathbf{1}^\top G\mathbf{1} - \frac{s}{n}(\mathbf{1}^\top G\mathbf{1})\gamma_s^*\right) = \min_{0 \leq s \leq k} \left(1 - \frac{s}{n}\gamma_s^*\right)\mathbf{1}^\top G\mathbf{1}.$$

Since the sequence $h(s) = \frac{s}{n}\gamma_s^*$ is non-decreasing, the term $(1 - h(s))$ is non-increasing. Therefore, the minimum value of $(1 - h(s))$ for $s \in \{0, \ldots, k\}$ occurs at $s = k$ (note $h(0) = 0$). Thus, $v_k^*(K, G) \leq \left(1 - \frac{k}{n}\gamma_k^*\right)\mathbf{1}^\top G\mathbf{1}$. The proof is complete. $\qquad\square$

**Implications for Different Values of $\rho_G$**    Proposition 4.3 provides insights into how the ratio $\rho_G$ affects the upper bound derived from the $\gamma\mathbf{1}$ analysis.

- If $\rho_G \leq 1$: In this case, $\gamma_k^* \geq 1$, hence $\frac{k}{n}\gamma_k^* \geq \frac{k}{n}$. We again recover the baseline linear decay guarantee: $v_k^*(K, G) \leq \left(1 - \frac{k}{n}\gamma_k^*\right)\mathbf{1}^\top G\mathbf{1} \leq \left(1 - \frac{k}{n}\right)\mathbf{1}^\top G\mathbf{1}$.

- If $\rho_G > 1$: Here, $\gamma_1^* \leq \gamma_k^* < 1$. The bound guarantees a decay rate of $\left(1 - \frac{k}{n}\gamma_k^*\right)$. Since $\gamma_k^* < 1$, this rate is slower than the linear $(1 - k/n)$ rate. The guaranteed rate is at least $\left(1 - \frac{k}{n}\gamma_1^*\right) = (1 - k/(n\rho_G))$. This highlights how significant cancellations ($\rho_G > 1$) slow down the guaranteed error reduction based on this specific analysis.

**Setting $\rho_G = 1/n$**   : The extremal setting where $\rho_G = 1/n$, which corresponds necessarily to $G = \alpha\mathbf{1}\mathbf{1}^\top$ for some $\alpha > 0$, is a specific instance discussed further in Appendix A.8. In this case $\gamma_s^* = 1/(\beta_s + (1 - \beta_s)/n)$, hence $\gamma_s^* = \frac{n}{s}$ for all $1 \leq s \leq n$. Thus, for $k \geq 1$, the upper estimate from Prop 4.3) becomes $(1 - \frac{k}{n}\gamma_k^*)\mathbf{1}^\top G\mathbf{1} = (1 - 1)\mathbf{1}^\top G\mathbf{1} = 0$. The zero optimal objective $v_k^*(K, G) = 0$ for all $1 \leq k \leq n$ implied by this is coherent. This occurs because for $s \geq 1$ fixed, the choice $y = \gamma_s^*\mathbf{1}$ (which equals $\frac{n}{s}\mathbf{1}$ in this case) leads to an average error of 0 in the proof derivation, meaning for this specific structure, no loss is incurred in the averaging argument used for proving Theorem 4.1]

**Generalizing the Ansatz**    The analysis for identifying solutions of the form $\gamma\mathbf{1}$ can be examined for solutions of the form $\gamma x$ for arbitrary $x$. Assuming $x$ is such that $\gamma_s^*(x)x \in K$ with $\gamma_s^*(x) := \frac{x^\top Gx}{x^\top G^{(s)}x}$, we get $w_s^*(K) \leq \phi_s(\gamma_s^*(x)x) = -(x^\top Gx)\gamma_s^*(x)$. This provides an alternative way to get bounds using different test vectors $x$.

### A.6   Computing the Upper Bound

Based on Theorem 4.1, we can outline Algorithm 1 for deriving an estimate the optimal value $v_k^*(K, G)$ of problem $\mathcal{P}_k(K, G)$.

---
**Algorithm 1** Estimate Upper Bound $u_k^*(K, G)$ for $k \in \{1, \ldots, n-1\}$
---
1: **procedure** ESTIMATEUPPERBOUND($G$, $K$, $k$)
2:     Compute $v_0 = \mathbf{1}^\top G\mathbf{1}$, $q = G\mathbf{1}$, $G_{\text{diag}} = \text{diag}(G)$
3:     Initialize $U = v_0$
4:     **for** $s = 1$ to $k$ **do**
5:         Construct the matrix $G^{(s)} = \beta_s G + (1 - \beta_s)G_{\text{diag}}$
6:         Consider the quadratic program $\mathcal{Q}_s(K)$: $\min_{y \in K}\left\{\phi_s(y) = y^\top G^{(s)}y - 2q^\top y\right\}$
7:         Solve $\mathcal{Q}_s(K)$ to obtain the optimal value $w_s^*(K)$
8:         Compute the upper bound candidate $u_s = v_0 + \frac{s}{n}w_s^*(K)$
9:         Update $U = \min(U, u_s)$
10:     **end for**
11:     **return** $U$                                    ▷ $U = \min_{0 \leq s \leq k} u_s$
12: **end procedure**

---

**Solving the Auxiliary Problems $\mathcal{Q}_s(K)$**    Since $G^{(s)}$ is positive definite for $s \in \{1, \ldots, n-1\}$ (assuming $G_{ii} > 0$), the program $\mathcal{Q}_s(K)$ admits a unique solution $y^{(s)}$ if $K$ is a non-empty, closed, convex set.

- For $K = \mathbb{R}^n$, the unique solution of equation Aux-QP is found by setting the gradient to zero: $2G^{(s)}y - 2G\mathbf{1} = 0$, which gives $y^{(s)} = (G^{(s)})^{-1}G\mathbf{1}$. The optimal value is $w_s^*(\mathbb{R}^n) = (y^{(s)})^\top G^{(s)}y^{(s)} - 2(G\mathbf{1})^\top y^{(s)} = (y^{(s)})^\top G\mathbf{1} - 2(G\mathbf{1})^\top y^{(s)} = -(G\mathbf{1})^\top y^{(s)}$.

- For $K = \mathbb{R}_+^n$ (non-negative orthant), the problem is a standard convex QP. The solution $y^{(s)}$ can be found using various QP solvers. It satisfies KKT conditions, potentially involving projection-like operations.

- For $K = [0, \xi]^n$ (box constraints), this is also a standard convex QP solvable by many algorithms. As discussed in Section 4, the main computational cost of Algorithm 1 lies in solving these $k$ convex QPs, which, while tractable, represents a higher complexity than typical approximation heuristics.

In cases where $K$ is convex and $w_s^*(K)$ is found (e.g., $w_s^*(K) = -(G\mathbf{1})^\top y^{(s)}$ for $K = \mathbb{R}^n$), the bound from Theorem 4.1 can be expressed using the optimal solution $y^{(s)}$ of the auxiliary problem:

$$v_k^*(K, G) \leq \min_{0 \leq s \leq k} \left( \mathbf{1}^\top G\mathbf{1} + \frac{s}{n} w_s^*(K) \right). \tag{14}$$

### A.6.1 APPLICATION TO MATRIX APPROXIMATION PROBLEMS

Using Proposition 4.2 (Eq. equation 6) for the binary selection case $K = \{0, 1\}^n$: Let $\eta_k = \frac{k}{n-1}$ for $k < n$ (and $\eta_n = 1$, although the bound is trivial for $k = n$). The bound on the optimal value $v_k^*(\{0, 1\}^n, G)$ is $\min(u_0, u_k)$ where $u_k = (1 - k/n)(\eta_k \operatorname{Tr}(G) + (1 - \eta_k)\mathbf{1}^\top G\mathbf{1})$ for $k < n$, and $u_0 = \mathbf{1}^\top G\mathbf{1}$. Assuming $\operatorname{Tr}(G) \leq \mathbf{1}^\top G\mathbf{1}$, the bound simplifies to $u_k$. If $\rho_G$ is such that $(1 - k/n)(\eta_k \rho_G + (1 - \eta_k)) \geq 1$ we only have the trivial bound $u_0$.

The bounds translates on the squared Frobenius norm error for specific problems:

- **Problem 1**: approximating $A\mathbf{1}$ using $k$ columns of $A$ (where $G = G_a = A^\top A$):

$$\min_{\substack{x \in \{0,1\}^n \\ \|x\|_0 \leq k}} \left\| A\mathbf{1} - \sum_{i=1}^n x_i a_i \right\|_2^2 \leq \left( 1 - \frac{k}{n} \right) \left( \eta_k \|A\|_F^2 + (1 - \eta_k)\|A\mathbf{1}\|_2^2 \right).$$

  The trivial bound $\|A\mathbf{1}\|_2^2$, and solution $x = \mathbf{0}$, is to be considered if the above is not better.

- **Problem 2**: approximating $AB^\top$ using $k$ outer products $a_i b_i^\top$ (here $G = G_{a,b} = G_a \odot G_b$):

$$\min_{\substack{x \in \{0,1\}^n \\ \|x\|_0 \leq k}} \left\| AB^\top - \sum_{i=1}^n x_i a_i b_i^\top \right\|_F^2 \leq \left( 1 - \frac{k}{n} \right) \left( \eta_k \sum_{i=1}^n (\|a_i\|_2 \|b_i\|_2)^2 + (1 - \eta_k)\|AB^\top\|_F^2 \right).$$

  Here $\operatorname{Tr}(G) = \sum_{i=1}^n (a_i^\top a_i)(b_i^\top b_i) = \sum_{i=1}^n \|a_i\|_2^2 \|b_i\|_2^2$, and $\mathbf{1}^\top G\mathbf{1} = \|AB^\top\|_F^2$. The trivial bound $\|AB^\top\|_F^2$ and solution $x = \mathbf{0}$ is to be considered if the above is not better.

- **Problem 3**: approximating $AA^\top$ using $k$ outer products $a_i a_i^\top$ (here $G = G_{a,a} = G_a \odot G_a$):

$$\min_{\substack{x \in \{0,1\}^n \\ \|x\|_0 \leq k}} \left\| AA^\top - \sum_{i=1}^n x_i a_i a_i^\top \right\|_F^2 \leq \left( 1 - \frac{k}{n} \right) \left( \eta_k \sum_{i=1}^n \|a_i\|_2^4 + (1 - \eta_k)\|AA^\top\|_F^2 \right).$$

  Here $\operatorname{Tr}(G) = \sum_{i=1}^n \|a_i\|_2^4$ and $\mathbf{1}^\top G\mathbf{1} = \|AA^\top\|_F^2$. As $\rho_{G_{a,a}} \leq 1$, this bound is guaranteed to be $\leq (1 - k/n)\|AA^\top\|_F^2$.

If we consider $x \in K$ with $[0, \max(1, \rho_{G,i}^{-1})]^n \subset K$ such as $K = \mathbb{R}^+$, associated upper estimates for the above three problem are

$$\left( 1 - \frac{k}{n} \gamma_{k,1}^* \right) \|A\mathbf{1}\|_2^2, \qquad \left( 1 - \frac{k}{n} \gamma_{k,2}^* \right) \|AB^\top\|_F^2, \qquad \left( 1 - \frac{k}{n} \gamma_{k,3}^* \right) \|AA^\top\|_F^2.$$

where $\gamma_{k,i}^* = (\beta_k + (1 - \beta_k)\rho_{G,i})^{-1}$ and $\rho_{G,i}$ is the structure ratio for each problem $i \in \{1, 2, 3\}$. In the third problem, we have $\rho_G \leq 1$, so that the decay $(1 - k/n)$ is guaranteed as already shown.

The identified decays

$$\left( 1 - \frac{k}{n} \right) ((1 - \eta_k) + \eta_k \rho_G), \qquad \text{and} \qquad 1 - \frac{k/n}{\beta_k + (1 - \beta_k)\rho_G}$$

in the unweighted/weighted approximation, become similar if $\rho_G \approx 1$.

### A.7 BOUND ON THE NORM OF OPTIMAL SOLUTIONS

Since $\mathcal{P}_k(\{0,1\}^n, G)$ is a restriction of the general problem $\mathcal{P}_k(K, G)$ to binary variables, optimal solutions for problem $\mathcal{P}_k(K, G)$ with constraint sets $K$ that include $\{0,1\}^n$ also satisfy the bounds derived for the binary case, specifically Proposition 4.2. This has implications for the norms of these solutions, as explored in the following corollary.

**Corollary A.2.** *Assume $G$ is non-singular (and thus positive definite, since it's PSD) and let $\lambda_{\min}(G) > 0$ be its smallest eigenvalue. Let $x^*$ be an optimal solution to problem $\mathcal{P}_k(K, G)$ for a constraint set $K$ such that $\{0,1\}^n \subset K$. Then $x^*$ satisfies*

$$\|x^*\|_2 \leq \sqrt{k} + \sqrt{\frac{\min(u_0, u_k)}{\lambda_{\min}(G)} - (n - k)} \tag{15}$$

*where $u_0 = \mathbf{1}^\top G \mathbf{1}$ and $u_k$ is given by Eq. equation 6 from Proposition 4.2:*

$$u_k = \left(1 - \frac{k}{n}\right)\left(\frac{k}{n-1}\operatorname{Tr}(G) + \left(1 - \frac{k}{n-1}\right)\mathbf{1}^\top G \mathbf{1}\right).$$

*Proof.* Let $\mathcal{S} \subset [n]$ be the support of $x^*$, with $|\mathcal{S}| \leq k$. We can assume $|\mathcal{S}| = k$ without loss of generality (if $|\mathcal{S}| < k$, we can add indices outside the support without changing $x^*$ or its norm). We denote by $x_{\mathcal{S}}^*$ the vector extracted from $x^*$ associated with indices in $\mathcal{S}$, and similarly $\mathbf{1}_{\mathcal{S}}$. Then $\|x^*\|_2 = \|x_{\mathcal{S}}^*\|_2$. By the triangle inequality, $\|x_{\mathcal{S}}^*\|_2 \leq \|\mathbf{1}_{\mathcal{S}}\|_2 + \|\mathbf{1}_{\mathcal{S}} - x_{\mathcal{S}}^*\|_2$. We have $\|\mathbf{1}_{\mathcal{S}}\|_2 = \sqrt{k}$. Also consider the vector $\mathbf{1} - x^*$. Its squared norm is $\|\mathbf{1} - x^*\|_2^2$. The components of $\mathbf{1} - x^*$ are $(1 - x_i^*)$ if $i \in \mathcal{S}$ and $1$ if $i \notin \mathcal{S}$. So, $\|\mathbf{1} - x^*\|_2^2 = \sum_{i \in \mathcal{S}}(1 - x_i^*)^2 + \sum_{i \notin \mathcal{S}} 1^2 = \|\mathbf{1}_{\mathcal{S}} - x_{\mathcal{S}}^*\|_2^2 + (n - k)$. Thus, $\|\mathbf{1}_{\mathcal{S}} - x_{\mathcal{S}}^*\|_2 = \sqrt{\|\mathbf{1} - x^*\|_2^2 - (n - k)}$. Combining these, we get: $\|x^*\|_2 \leq \sqrt{k} + \sqrt{\|\mathbf{1} - x^*\|_2^2 - (n - k)}$. Since $G$ is positive definite, its smallest eigenvalue $\lambda_{\min}(G)$ is positive. We have the standard inequality relating the quadratic form to the Euclidean norm: $(\mathbf{1} - x^*)^\top G(\mathbf{1} - x^*) \geq \lambda_{\min}(G)\|\mathbf{1} - x^*\|_2^2$. The objective value for the optimal solution $x^*$ is $v_k^*(K, G) = (\mathbf{1} - x^*)^\top G(\mathbf{1} - x^*)$. Since $\{0,1\}^n \subset K$, the optimal value $v_k^*(K, G)$ must be less than or equal to the optimal value for the binary case, $v_k^*(\{0,1\}^n, G)$. From Proposition 4.2, we know $v_k^*(\{0,1\}^n, G) \leq \min(u_0, u_k)$, where $u_k$ is given by equation 6. Therefore, $\lambda_{\min}(G)\|\mathbf{1} - x^*\|_2^2 \leq v_k^*(K, G) \leq \min(u_0, u_k)$. This implies $\|\mathbf{1} - x^*\|_2^2 \leq \frac{\min(u_0, u_k)}{\lambda_{\min}(G)}$. Substituting this into the inequality for $\|x^*\|_2$: $\|x^*\|_2 \leq \sqrt{k} + \sqrt{\frac{\min(u_0, u_k)}{\lambda_{\min}(G)} - (n - k)}$. The expression under the square root must be non-negative for the bound to be meaningful. The proof is complete. $\square$

### A.8 ANALYSIS FOR SPECIAL GRAM MATRIX STRUCTURES

#### A.8.1 ANALYSIS IN SPECIAL SETTINGS

This section consolidates discussions regarding specific structures of the matrix $G$.

**Diagonal G (Orthogonal Terms)** Consider the case where $G$ is diagonal, $G = \operatorname{diag}(G_{11}, \ldots, G_{nn})$. This corresponds to the situation where the rank-one terms $a_i b_i^\top$ are mutually orthogonal under the Frobenius inner product, i.e., $\langle a_i b_i^\top, a_j b_j^\top \rangle_F = G_{ij} = 0$ for $i \neq j$. In this case, $\operatorname{Tr}(G) = \sum_{i=1}^n G_{ii}$ and $\mathbf{1}^\top G \mathbf{1} = \sum_{i=1}^n G_{ii}$, so $\rho_G = \frac{\operatorname{Tr}(G)}{\mathbf{1}^\top G \mathbf{1}} = 1$. The interpolated matrix $G^{(s)} = \beta_s G + (1 - \beta_s)\operatorname{diag}(G_{11}, \ldots, G_{nn}) = \beta_s G + (1 - \beta_s)G = G$ for all $s$. The auxiliary problem $\mathcal{Q}_s(K)$ becomes $\min_{y \in K} y^\top G y - 2(G\mathbf{1})^\top y$. Let's examine the bounds:

- **Prop. 4.2 (Binary Case):** With $\rho_G = 1$, equation 6 gives

$$u_k = (1 - k/n)(1 + \frac{k}{n-1}(1-1))\mathbf{1}^\top G \mathbf{1} = (1 - k/n)\mathbf{1}^\top G \mathbf{1}.$$

  The bound is $v_k^* \leq (1 - k/n)\operatorname{Tr}(G)$. This is sharp; the optimal strategy is to select the $k$ indices corresponding to the largest $G_{ii}$ values, setting $x_i = 1$ for these. The error is exactly the sum of the $n - k$ smallest $G_{ii}$ values. If all $G_{ii}$ are equal (say, to $G_{11}$), the error is $(n - k)G_{11} = (1 - k/n)nG_{11} = (1 - k/n)\operatorname{Tr}(G)$.

- **Prop. 4.3 (Scaled Identity):** With $\rho_G = 1$, we have $\gamma_k^* = 1/(\beta_k + (1 - \beta_k) \cdot 1) = 1$. The bound equation 7 becomes $v_k^*(K, G) \leq (1 - k/n)\mathbf{1}^\top G\mathbf{1} = (1 - k/n)\operatorname{Tr}(G)$. This matches the binary case bound.

- **Theorem 4.1 (General Bound):** If we solve $\mathcal{Q}_s(K)$ for $K = \mathbb{R}^n$, the solution is $y^{(s)} = (G^{(s)})^{-1}G\mathbf{1} = G^{-1}G\mathbf{1} = \mathbf{1}$ (since $G$ is diagonal and assumed invertible). The optimal value is $w_s^*(\mathbb{R}^n) = -(G\mathbf{1})^\top \mathbf{1} = -\mathbf{1}^\top G\mathbf{1} = -\operatorname{Tr}(G)$. The bound becomes $u_s = \operatorname{Tr}(G) + (s/n)(-\operatorname{Tr}(G)) = (1 - s/n)\operatorname{Tr}(G)$. Minimizing over $s \in \{1, \ldots, k\}$ gives $u_k = (1 - k/n)\operatorname{Tr}(G)$.

All approaches consistently yield the $(1 - k/n)\operatorname{Tr}(G)$ bound for diagonal $G$.

**Rank-One G (Extremal Correlation)**   Consider the case where $G$ has rank one, $G = gg^\top$ for some vector $g \in \mathbb{R}^n$. We assume $G \neq 0$. Then $\operatorname{Tr}(G) = \operatorname{Tr}(gg^\top) = g^\top g = \|g\|_2^2$. And $\mathbf{1}^\top G\mathbf{1} = \mathbf{1}^\top gg^\top \mathbf{1} = (g^\top \mathbf{1})^2 = (\sum g_i)^2$. The structure ratio is $\rho_G = \frac{\operatorname{Tr}(G)}{\mathbf{1}^\top G\mathbf{1}} = \frac{\|g\|_2^2}{(g^\top \mathbf{1})^2}$. By the Cauchy-Schwarz inequality, $(g^\top \mathbf{1})^2 \leq \|\mathbf{1}\|_2^2\|g\|_2^2 = n\|g\|_2^2$, so $\rho_G \geq 1/n$. Equality holds if and only if $g$ is proportional to $\mathbf{1}$.

If $g = \alpha\mathbf{1}$ for some $\alpha \neq 0$ (i.e., $G = \alpha^2\mathbf{1}\mathbf{1}^\top$). Here $\rho_G = \frac{\|\alpha\mathbf{1}\|_2^2}{(\alpha\mathbf{1}^\top\mathbf{1})^2} = \frac{\alpha^2 n}{(\alpha n)^2} = \frac{\alpha^2 n}{\alpha^2 n^2} = 1/n$. Looking at Proposition 4.2: As shown previously (Eq. equation 12), the bound becomes $v_k^* \leq (1 - k/n)^2\mathbf{1}^\top G\mathbf{1}$.

**Improved Rates for Constant Diagonal G**

**Proposition A.3.** *Assume that* $\operatorname{diag}(G) = \alpha\mathbf{1}$ *for some* $\alpha > 0$ *(i.e.,* $G_{ii} = \alpha$ *for all* $i$*). Let* $r = \operatorname{rank}(G)$ *and let* $G = \sum_{i=1}^r \lambda_i u_i u_i^\top$ *be its reduced eigenvalue decomposition. Define* $c = \frac{1}{n}\left(\sum_{i=1}^r \langle \mathbf{1}, u_i\rangle^2\right) = \frac{\|P_{\operatorname{range(G)}}\mathbf{1}\|_2^2}{n} \in [0, 1]$, *where* $P_{\operatorname{range(G)}}$ *is the orthogonal projector onto the range of* $G$*. Define* $\widetilde{\gamma}_s^*$ *as:*

$$\widetilde{\gamma}_s^* = \frac{1}{\beta_s + (1 - \beta_s)c\rho_G}, \quad 1 \leq s \leq n. \tag{16}$$

*For* $K = \mathbb{R}^n$*, the optimal value* $v_k^*(K, G)$ *is bounded according to*

$$v_k^*(\mathbb{R}^n, G) \leq \left(1 - \frac{k}{n}\widetilde{\gamma}_k^*\right)\mathbf{1}^\top G\mathbf{1}, \quad 1 \leq k \leq n. \tag{17}$$

*Proof.* We consider the vector $x = P_{\operatorname{range(G)}}\mathbf{1} = \sum_{i=1}^r \langle \mathbf{1}, u_i\rangle u_i$, the projection of $\mathbf{1}$ onto the range of $G$. We evaluate the objective $\phi_s(y)$ from Theorem 4.1 using the ansatz $y = \gamma x$. We need $x^\top Gx$ and $x^\top G^{(s)}x$. First,

$$x^\top Gx = \sum_{i=1}^r \lambda_i\langle \mathbf{1}, u_i\rangle^2 = \mathbf{1}^\top G\mathbf{1}$$

Now consider $G^{(s)} = \beta_s G + (1 - \beta_s)\operatorname{diag}(G) = \beta_s G + (1 - \beta_s)\alpha I_n$. Then

$$x^\top G^{(s)}x = \beta_s x^\top Gx + (1 - \beta_s)\alpha\|x\|^2 = \beta_s\mathbf{1}^\top G\mathbf{1} + (1 - \beta_s)n\alpha\frac{\|x\|^2}{n}.$$

Also, $\operatorname{Tr}(G) = \sum G_{ii} = n\alpha$ thus the structure ratio is $\rho_G = \frac{\operatorname{Tr}(G)}{\mathbf{1}^\top G\mathbf{1}} = \frac{n\alpha}{\mathbf{1}^\top G\mathbf{1}}$. Using $c = \|x\|^2/n$ and substituting these into the expression for $x^\top G^{(s)}x$:

$$x^\top G^{(s)}x = (\beta_s + (1 - \beta_s)c\rho_G)\mathbf{1}^\top G\mathbf{1}.$$

The objective function in Theorem 4.1 is $\phi_s(y) = y^\top G^{(s)}y - 2(G\mathbf{1})^\top y$. For $y = \gamma x$, this becomes

$$\phi_s(\gamma x) = \gamma^2(x^\top G^{(s)}x) - 2\gamma(G\mathbf{1})^\top x = \gamma^2(x^\top G^{(s)}x) - 2\gamma\mathbf{1}^\top G\mathbf{1}.$$

This quadratic in $\gamma$ is minimized at $\gamma_{unc} = \frac{\mathbf{1}^\top G\mathbf{1}}{x^\top G^{(s)}x}$ and the optimal value is $-\gamma_{unc}\mathbf{1}^\top G\mathbf{1}$. Substituting the expression for $x^\top G^{(s)}x$ we have:

$$\gamma_{unc} = \frac{\mathbf{1}^\top G\mathbf{1}}{(\mathbf{1}^\top G\mathbf{1})[\beta_s + (1 - \beta_s)c\rho_G]} = \frac{1}{\beta_s + (1 - \beta_s)c\rho_G} = \widetilde{\gamma}_s^*.$$

Since $y = \widetilde{\gamma}_s^* x$ is feasible for $K = \mathbb{R}^n$ (as $K$ imposes no constraints), the true minimum $w_s^*(\mathbb{R}^n)$ must be less than or equal to this value: $w_s^*(\mathbb{R}^n) \leq -(\mathbf{1}^\top G\mathbf{1})\widetilde{\gamma}_s^*$. Plugging this upper bound on $w_s^*(\mathbb{R}^n)$ into the bound from Theorem 4.1:

$$v_k^*(\mathbb{R}^n, G) \leq \min_{0 \leq s \leq k} \left( \mathbf{1}^\top G\mathbf{1} + \frac{s}{n} w_s^*(\mathbb{R}^n) \right)$$

$$\leq \min_{0 \leq s \leq k} \left( \mathbf{1}^\top G\mathbf{1} - \frac{s}{n}(\mathbf{1}^\top G\mathbf{1})\widetilde{\gamma}_s^* \right)$$

$$= \min_{0 \leq s \leq k} \left( 1 - \frac{s}{n}\widetilde{\gamma}_s^* \right) \mathbf{1}^\top G\mathbf{1}.$$

Assuming the function $f(s) = \frac{s}{n}\widetilde{\gamma}_s^*$ is non-decreasing for $s \in \{1, \ldots, n\}$ (which holds under typical conditions for $\beta_s$), the minimum occurs at $s = k$. Therefore, $v_k^*(\mathbb{R}^n, G) \leq \left( 1 - \frac{k}{n}\widetilde{\gamma}_k^* \right) \mathbf{1}^\top G\mathbf{1}$. The proof is complete. $\qquad\square$

Note that in this specific setting (constant diagonal), we only require $c\rho_G \leq 1$, that is $\mathrm{Tr}(G) \leq (\mathbf{1}^\top G\mathbf{1})/c = n(\mathbf{1}^\top G\mathbf{1})/\|P_{\mathrm{range(G)}}\mathbf{1}\|_2^2$, in order to ensure $\widetilde{\gamma}_k^* \geq 1$. If $\widetilde{\gamma}_k^* \geq 1$, then the bound implies a linear decay rate $v_k^*(\mathbb{R}^n, G) \leq \left( 1 - \frac{k}{n} \right) \mathbf{1}^\top G\mathbf{1}$.

**Selected Upper bounds on AMM:** A recapitulative table is provided in Table 3

Table 3: Selected Upper Bounds on Approximate Matrix Multiplication Error ($k$ = samples/sketch dimension/terms).

| Method/Setting | Upper Bound on Error | Ref. |
|---|---|---|
| *Bounds on Frobenius Norm Error $E[\|M - \widetilde{C}\|_F^2]$ or related ($M = AB^\top$)* | | |
| AMM (Optimal Sampling $p_i \propto G_{ii}$) | $\frac{1}{k}\left(\sum_{i=1}^n \sqrt{G_{ii}}\right)^2 - \frac{1}{k}\|M\|_F^2$ | (Drineas et al., 2006a) |
| | $\leq \frac{1}{k}\mathrm{Tr}(G)(\text{factor})^2 - \frac{1}{k}\|M\|_F^2$ | |
| AMM (Uniform Sampling $p_i = 1/n$) | $\frac{n}{k}\mathrm{Tr}(G) - \frac{1}{k}\|M\|_F^2$ | (Drineas et al., 2006a) |
| Randomized Sketching (e.g., CountSketch, SRHT, Gaussian) | $\approx \frac{1}{k}\|A\|_F^2\|B\|_F^2$ | (Sarlos, 2006; Clarkson & Woodruff, 2017; Cohen et al., 2023) |
| | (Ignoring structure $\|M\|_F^2, \rho_G$) | |
| *Bounds on Frobenius Norm Error $E[\|M - \widetilde{C}\|_F^2]$ or related ($M = AA^\top$)* | | |
| AMM (Optimal Sampling, Gram $p_i \propto \|A_i\|^4$) | $\frac{1}{k}\sum_{i=1}^n \|A_i\|^4 = \frac{1}{k}\mathrm{Tr}(G_{a,a})$ | (Drineas et al., 2006a) |
| Nyström (Uniform Sampling, Gram) | $\approx \|M - M_k\|_F^2 + \frac{n}{k}\sum_{i=k+1}^n \sigma_i(M)^2$ | (Drineas & Mahoney, 2005; Gittens & Mahoney, 2013) |
| Deterministic Sketching (Freq. Directions, Gram) | $\|M - \widetilde{C}\|_F^2 \leq \frac{1}{k+1}\|A\|_F^4$ | (Ghashami et al., 2016) |
| *Bounds on Spectral Norm Error $\|M - \widetilde{C}\|$ or related (w.h.p.)* | | |
| Random Projection (OSE-based, $AB^\top$) | $\leq \epsilon\|A\|\|B\|$ | (Cohen et al., 2016) |
| | (Requires $k \propto (\mathrm{sr}(A) + \mathrm{sr}(B))/\epsilon^2$) | |
| Leverage Scores (Optimal Sampling, $AB^\top$) | $\leq \frac{C}{\sqrt{k}}\min(\|A\|_F\|B\|, \|A\|\|B\|_F)$ | (Cohen et al., 2016) |
| *Other Bounds* | | |
| Greedy Selection (a posteriori, $AB^\top$) | $\|M - \sum_{j\in J} w_j A_j B_j^\top\|_F^2$ | (Belabbas & Wolfe, 2008; Tropp & Gilbert, 2007) |
| | (Error depends on specific choice $J$, weights $w_j$) | |
| **(Upper Bound on Optimal Error, $AB^\top$)** | $v_k^*(K, G) \leq u_k^*(K, G)$ | **This Paper** |
| | ($u_k^*(K,G)$ computable via QP, depends explicitly on $G$) | |

# B  APPENDIX: ADDITIONAL EXPERIMENTAL RESULTS AND MATRIX GENERATION DETAILS

This appendix provides detailed methodologies for generating the diverse matrix types used in our experiments, all configured with dimensions $n$ (columns), $m$ (rows for matrix $A$), and $p$ (rows for matrix $B$). We also present a comprehensive set of plots corresponding to these matrix types, illustrating the performance of our proposed bounds alongside other algorithms. The relative squared approximation error is defined as $\|AB^T - \tilde{A}\tilde{B}^T\|_F^2/\|AB^T\|_F^2$, plotted against the sampling ratio $k/n$.

## B.1  MATRIX GENERATION METHODOLOGIES

The following subsections detail the generation process for each matrix type.

**i.i.d. Gaussian Matrices**  Matrices $A \in \mathbb{R}^{m \times n}$ and $B \in \mathbb{R}^{p \times n}$ are generated with all entries drawn independently from a standard Gaussian distribution $\mathcal{N}(0, 1)$. This fundamental matrix type is produced using our `generate_matrices_gaussian_cancellation` function by setting the `cancel_fraction = 0.0` and `noise_level = 0.0`. This setup serves as a common baseline, characterized by no specific induced correlation or cancellation structures.

**Uniform [-1, 1] Matrices**  For this type, entries for both matrices $A$ and $B$ are drawn independently and uniformly at random from the interval $[-1, 1]$. This is implemented using the `generate_matrices_uniform` function with parameters `low = -1.0` and `high = 1.0`. These matrices allow for the evaluation of algorithms on non-Gaussian, bounded data.

**Row Orthogonal Matrices**  To generate matrices with orthogonal rows, we first create random Gaussian matrices of appropriate dimensions and then apply QR decomposition to obtain orthogonal bases. Specifically, for matrix $A \in \mathbb{R}^{m \times n}$, we generate a random Gaussian matrix $X \in \mathbb{R}^{n \times m}$, compute its QR decomposition $X = QR$, and set $A = Q^T$. Similarly for $B$, we generate a random Gaussian matrix $Y \in \mathbb{R}^{n \times p}$, compute its QR decomposition, and set $B = Q^T$. This process ensures that $AA^T = I_m$ and $BB^T = I_p$, meaning the rows of both matrices form orthonormal bases. This is implemented using the `generate_matrices_row_orthogonal` function.

**Gaussian Matrices with 10% Column Cancellation**  Matrices $A$ and $B$ are initially filled with i.i.d. Gaussian entries $\mathcal{N}(0, 1)$. A cancellation effect is then introduced for 10% of the columns (500 columns for $n = 5000$), selected uniformly at random. For each selected column index $j$, the $j$-th column of $B$ is made proportional to the negative of the corresponding column in $A$ (for the minimum number of rows shared between $A$ and $B$), creating a cancellation effect in the product $AB^T$. After this structural modification, a small amount of Gaussian noise, scaled to 5% of the standard deviation of the original matrix entries (`noise_level = 0.05`), is added to all elements of both $A$ and $B$. This type is generated using the `generate_matrices_gaussian_cancellation` function with `cancel_fraction = 0.1`.

**Matrices with 10% Repeated Columns**  For this type, we first generate matrices $A$ and $B$ with i.i.d. Gaussian entries. We then randomly select 10% of the columns to be replaced with duplicates of other randomly selected columns. Specifically, we keep 90% of the columns as unique, and the remaining 10% are set as exact copies of randomly selected columns from the unique set. After this structural modification, a small amount of Gaussian noise, scaled to 1% of the Frobenius norm of the matrices, is added to all elements. This is implemented using the `generate_matrices_repeated_cols` function with parameters `repeat_frac = 0.1` and `noise_ratio = 0.01`.

**NonLinear Tanh(Gaussian) Matrices**  To generate these matrices, intermediate matrices $A'$ and $B'$ are first populated with i.i.d. standard Gaussian entries $\mathcal{N}(0, 1)$. The final matrices $A$ and $B$ are then obtained by applying the hyperbolic tangent function (tanh) element-wise to $A'$ and $B'$ respectively, i.e., $A_{ij} = \tanh(A'_{ij})$ and $B_{ij} = \tanh(B'_{ij})$. This transformation is handled by the `generate_matrices_nonlinear` function with `base_type='gaussian'` and `func=np.tanh`. This process introduces significant non-linearity, resulting in matrix entries

bounded within $(-1, 1)$ but with a distribution markedly different from simple uniform or Gaussian distributions, challenging assumptions of linearity or specific distributional forms.

The following figures present the experimental results for the matrix types detailed above.

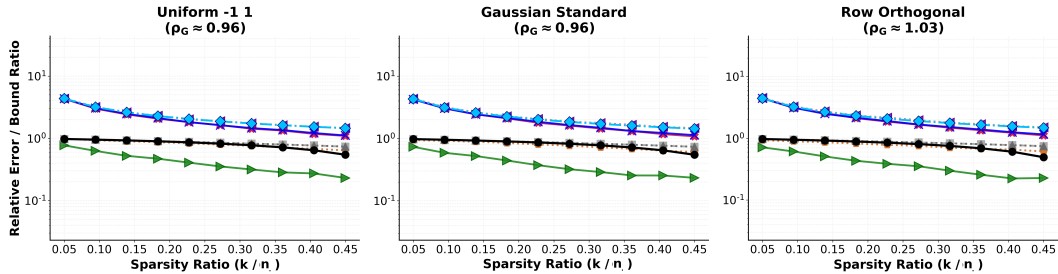

Figure 5: Performance comparison for Uniform, Gaussian, and Row Orthogonal matrix types ($n = 5000, m = 50, p = 30$). The Scaled Id Bound (Proposition 4.3) consistently tracks the Aux QP Bound (Theorem 4.1). Plots show relative error vs. sampling ratio $k/n$.

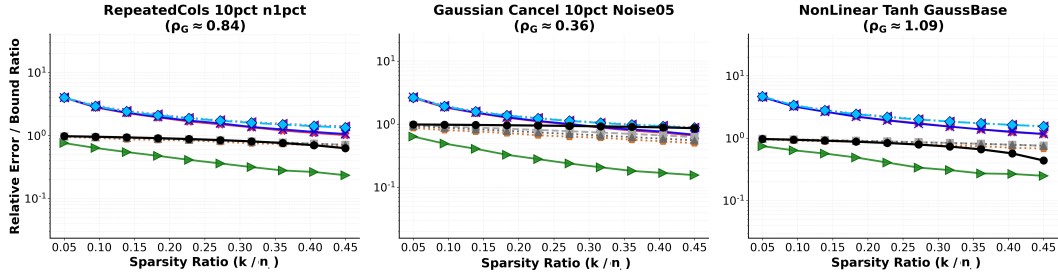

Figure 6: Performance comparison for Repeated Columns, RBF Features, and NonLinear Tanh matrix types ($n = 5000, m = 50, p = 30$). Our bounds remain tight across these diverse matrix structures, while OMP consistently achieves the lowest error. Plots show relative error vs. sampling ratio $k/n$.

### B.2  ORTHOGONAL MATCHING PURSUIT (OMP) FOR MATRIX PRODUCT APPROXIMATION

To demonstrate the practical achievability of our theoretical bounds, we adapt Orthogonal Matching Pursuit (OMP) to the matrix product approximation problem. OMP is a greedy iterative algorithm that selects columns one by one to minimize the approximation error.

#### B.2.1  PROBLEM FORMULATION

Given matrices $A \in \mathbb{R}^{m \times n}$ and $B \in \mathbb{R}^{p \times n}$, our goal is to select a subset of $k$ columns (indexed by set $S$) to minimize:

$$\min_{S \subset [n], |S|=k} \|AB^T - A_S X B_S^T\|_F^2 \tag{18}$$

where $A_S$ and $B_S$ are submatrices formed by the columns indexed by $S$, and $X$ is an optimal weight matrix.

#### B.2.2  ALGORITHM DESCRIPTION

The OMP algorithm for matrix product approximation is presented in Algorithm 2.

---

**Algorithm 2** Orthogonal Matching Pursuit for Matrix Product Approximation

---

**Require:** Matrices $A \in \mathbb{R}^{m \times n}$, $B \in \mathbb{R}^{p \times n}$, target number of columns $k$
**Ensure:** Set of selected column indices $S$ with $|S| = k$
 1: Initialize residual $R_0 \leftarrow AB^T$
 2: Initialize selected indices $S_0 \leftarrow \emptyset$
 3: **for** $j = 1$ to $k$ **do**
 4:      **Matching Step:** Find column index

$$s_j \leftarrow \arg\max_{l \notin S_{j-1}} |\langle R_{j-1}, a_l b_l^T \rangle_F| = |\text{Tr}(R_{j-1}^T a_l b_l^T)| \qquad (19)$$

     where $a_l$ and $b_l$ are the $l$-th columns of $A$ and $B$ respectively
 5:      Update selected indices: $S_j \leftarrow S_{j-1} \cup \{s_j\}$
 6:      **Optimal Weight Computation:** Solve for optimal weight matrix $X_j^*$:

$$X_j^* \leftarrow \arg\min_{X \in \mathbb{R}^{j \times j}} \|AB^T - A_{S_j} X B_{S_j}^T\|_F^2 \qquad (20)$$

 7:      **Projection Step:** Update approximation using optimal weights
 8:      Update residual: $R_j \leftarrow AB^T - A_{S_j} X_j^* B_{S_j}^T$
 9: **end for**
10: Return $S_k$, $X_k^*$

---

### B.2.3 IMPLEMENTATION DETAILS

The optimal weight matrix $X_j^*$ in step 6 can be computed by solving the least squares problem:

$$X_j^* = (A_{S_j}^T A_{S_j})^{-1} (A_{S_j}^T AB^T B_{S_j})(B_{S_j}^T B_{S_j})^{-1} \qquad (21)$$

This formulation ensures that at each iteration, we not only select the most promising column but also optimally reweight all previously selected columns to minimize the approximation error. This is crucial for achieving the best possible approximation with the selected columns.

The relative squared approximation error reported for OMP at step $k$ is:

$$\text{Error}_k = \frac{\|AB^T - A_{S_k} X_k^* B_{S_k}^T\|_F^2}{\|AB^T\|_F^2} \qquad (22)$$

### B.3 ANALYSIS OF OMP PERFORMANCE AND NUMERICAL EFFECTS AT HIGH $\rho_G$

As observed in our experiments, particularly in Figure 5, OMP generally exhibits strong empirical performance that closely tracks our theoretical bounds. However, we note some minor non-monotonic behavior in its error curve, especially for small $n$ (e.g., $n = 30$) and high $\rho_G$ values. This section explains the underlying causes of this phenomenon.

### B.3.1 CHALLENGES IN HIGH-$\rho_G$ REGIMES

To construct matrices with high $\rho_G$ values, our generation process (detailed in Section B.1) creates specific structural patterns that can introduce numerical challenges:

- **Induced Cancellations:** We create pairs of columns with properties such as $A_j \approx cA_i$ and $B_j \approx -c'B_i$, making off-diagonal terms in $(A^T A) \odot (B^T B)$ significant and potentially negative.

- **Near-Collinearity:** Some columns become nearly collinear, creating ill-conditioned sub-problems during the OMP projection step.

- **Large Dynamic Range:** Matrix entries may span a wide range of magnitudes, exacerbating floating-point precision issues.

### B.3.2 NUMERICAL EFFECTS ON OMP PERFORMANCE

These characteristics lead to several numerical challenges that affect OMP's performance:

1. **Ill-Conditioned Subproblems:** When OMP selects columns that are part of an induced cancellation structure, computing the optimal weights may involve inverting ill-conditioned matrices $(A_{S_j}^T A_{S_j})$ and $(B_{S_j}^T B_{S_j})$, leading to numerical instability.

2. **Residual Calculation Sensitivity:** Computing $R_j = AB^T - A_{S_j} X_j^* B_{S_j}^T$ involves subtracting two potentially large, nearly equal matrices. This subtraction can suffer from catastrophic cancellation in floating-point arithmetic.

3. **Error Propagation:** The greedy nature of OMP means that small numerical errors in early iterations can lead to suboptimal column selections, with effects that compound in subsequent iterations.

When $n$ is small (as in our $n = 30$ experiments), these numerical effects have a more pronounced impact:

- The limited pool of columns means fewer alternatives when a numerically-influenced suboptimal choice is made

- The relative impact of each column selection is greater

- The error curve becomes more sensitive to individual column selections

Despite these numerical challenges, OMP's overall performance remains strong and closely tracks our theoretical bounds, confirming that our bounds are not only theoretically sound but also practically achievable. The minor non-monotonicity observed in high-$\rho_G$ regimes is primarily a numerical artifact rather than a fundamental limitation of the algorithm.

### B.4 GENERATION OF MATRICES WITH CONTROLLED $\rho_G$

To systematically evaluate algorithm performance under varying matrix structures, we define the structural metric $\rho_G$ for matrices $A \in \mathbb{R}^{m \times n}$ and $B \in \mathbb{R}^{p \times n}$ as:

$$\rho_G(A, B) = \frac{\text{Tr}((A^T A) \odot (B^T B))}{\sum_{i,j}((A^T A) \odot (B^T B))_{ij}} = \frac{\sum_{l=1}^n (A^T A)_{ll}(B^T B)_{ll}}{\sum_{i=1}^n \sum_{j=1}^n (A^T A)_{ij}(B^T B)_{ij}} \tag{23}$$

where $\odot$ denotes the element-wise (Hadamard) product.

Our matrix generation procedure employs two primary mechanisms to control $\rho_G$:

**Induced Column Cancellations** For a specified number of column pairs, we create structural dependencies that influence the off-diagonal elements of $(A^T A) \odot (B^T B)$. Specifically, for selected pairs of columns $(i, j)$:

- We set $A_j \approx c \cdot A_i$ (where $c$ is close to 1)

- We set $B_j \approx -c' \cdot B_i$ (where $c'$ is close to 1)

This creates negative products in the off-diagonal terms: if $A_j \approx A_i$ and $B_j \approx -B_i$, then $\langle A_i, A_j \rangle \approx \|A_i\|^2$ and $\langle B_i, B_j \rangle \approx -\|B_i\|^2$, making their product negative. This reduces the denominator of $\rho_G$ relative to the numerator, increasing $\rho_G$.

The number of cancellation pairs is adjusted based on the target $\rho_G$:

- Low $\rho_G$ (0.5-1.0): Few to no cancellation pairs

- Medium $\rho_G$ (2.0-10.0): 10-25% of columns involved in pairs

- High $\rho_G$ (>10.0): 25-50% of columns involved in pairs

**Controlled Noise Injection**    We add Gaussian noise scaled to a percentage (typically 0-15%) of the standard deviation of the original matrix entries. This noise:

- Prevents perfect cancellations that could make the denominator of $\rho_G$ too small
- Introduces variability in the distribution of values in $A^T A$ and $B^T B$
- Helps achieve very high or very low target $\rho_G$ values by breaking symmetries

Our generation algorithm iteratively tries different combinations of cancellation pairs and noise levels until it finds matrices with $\rho_G$ within a specified tolerance of the target (typically ±25-30%). This approach allows us to generate datasets spanning a controlled range of $\rho_G$ values for comprehensive algorithm evaluation.

### B.5    ALGORITHMS AND BASELINES

We implemented and compared the following algorithms and bounds in our experiments:

**Optimal Error** ($v_k^*$)    The true minimum squared Frobenius norm error achievable by selecting the best $k$ columns:

$$v_k^* = \min_{S:|S|=k} \|AB^T - A_S X B_S^T\|_F^2 \tag{24}$$

This is computed via exhaustive search for small $n$, with $X$ being a weight matrix.

**Our Proposed Bounds**

- **Aux QP Bound (Theorem 4.1)**: Derived from solving a Quadratic Program, providing a tight upper bound on $v_k^*$. Implemented using CVXPY Agrawal et al. (2018); Diamond & Boyd (2016).
- **Scaled Id Bound (Proposition 4.3)**: An analytical approximation to the QP bound that is computationally more efficient while maintaining tightness.

**Algorithms**

- **Orthogonal Matching Pursuit (OMP)**: The greedy algorithm described in Section B.2, which iteratively selects columns to minimize the residual error. Based on the approach in Tropp & Gilbert (2007).
- **Optimal Sampling**: Columns are sampled proportionally to their norms. We compare against the theoretical bound from Drineas et al. (2006a).
- **Gaussian Projection**: A sketching method where matrices are projected onto random Gaussian matrices. Based on the approach in Sarlos (2006).
- **CountSketch**: A sketching technique based on hashing Charikar et al. (2004).
- **Subsampled Randomized Hadamard Transform (SRHT)**: A fast sketching method using Hadamard transforms TROPP (2011) (Boutsidis & Gittens (2013) for specific implementation).

For each algorithm, we report the relative squared approximation error:

$$\text{Error} = \frac{\|AB^T - \tilde{A}\tilde{B}^T\|_F^2}{\|AB^T\|_F^2} \tag{25}$$

where $\tilde{A}$ and $\tilde{B}$ are the approximations produced by the respective algorithm.

