# OpenReview forum: "Unveiling the Hidden Structure: Tight Bounds for Matrix Multiplication Approximation via Convex Optimization"
_ICLR.cc/2026/Conference — Submitted to ICLR 2026_

### Official Review · Reviewer_hsCA · 2025-10-16

**Soundness:** 3
**Presentation:** 3
**Contribution:** 2
**Rating:** 6
**Confidence:** 3

**Summary:**

This paper proposes a new complexity measure for approximate matrix multiplication in the Frobenius norm. Let $G=(A^\top A)\circ (B^\top B)$, the measure $\rho_G=\frac{{\rm Tr}(G)}{{\bf 1}^T G {\bf 1}}$. Moreover, it studies the algorithmic question of finding an optimal weight vector $x\in K$ with sparsity at most $k$ such that $\\|AB^\top-AXB^\top \\|_F$ is minimized, where $X$ is the diagonal matrix of vector $x$. Authors formulate it as a quadratic program with $\ell_0$ constraint, which is NP-hard to solve. To overcome it, they propose to solve a convex quadratic program whose optimal value gives an upper bound on the optimal value of the sparse quadratic program. Several other simpler analytical upper bounds for particular convex sets $K$ are also studied. Finally, they empirically verify the gap between several popular AMM algorithms, including orthogonal matching pursuit, optimal sampling, and several oblivious subspace embedding algorithms and the optimal bound, upper bound of it, and several proxies, showing that OMP is much closer to the optimal than other approaches.

**Strengths:**

* The proposed complexity measure is a more robust and unified estimate of the product $AB^\top$, as it goes beyond estimate that only depends on norms, e.g., $\\|A\\|_F \\|B\\|_F$, and it captures how outer products $a_ib_i^\top$ interact with each other.

* The corresponding algorithmic problem, i.e., finding an optimal $k$-sparse weight vector also fits quite naturally, as various sampling methods have been studied before with various guarantees. This offers new perspective on studying traditional sampling algorithms such as leverage score sampling and row norm based sampling.

* The study of the sparse convex program and its relaxations are quite thorough, a tractable convex quadratic relaxation is proposed and analyzed, and when the constraint sets are structured, better and simpler analytical bounds are studied. These bounds have the advantage that one can very quickly estimate them without solving the convex quadratic relaxation.

* Experiments performed, which in my opinion, are quite insightful, as they show the fundamental limits of sketching and sampling based methods when $\rho_G$ is large. In addition, they show that the most input-dependent algorithm OMP unsurprisingly gives the best error, albeit it is the slowest algorithm.

**Weaknesses:**

* The theoretical analysis in this paper merely provides an upper bound on the optimal **value** of the sparse convex quadratic program, a much more interesting and practical question is how to efficiently compute an approximately optimal **solution**. For example, is there a rounding scheme for the convex quadratic program that serves as an upper bound on the value, that could also lead to a $k$-sparse approximate weighting? While the analysis on the optimal value is quite insightful in benchmarking different AMM algorithms, it falls short on converting the framework into a useful algorithm.

* The convex program studied in this paper has a significant gap to oblivious sketching based algorithms, as these algorithms require possibly to linear combine all the columns, meaning that one cannot simply write it as $\sum_i x_i a_ib_i^\top$, I believe the formulation studied in this paper applies only to sampling algorithms. While one could argue that the performance of these sketching algorithms could still be compared, this should be clarified.

* The implication and applicability of the optimal AMM should be further elaborated. In many modern day ML applications, in particular deep learning settings, matrices $A$ and $B$ are changing constantly, prior AMM tools such as sampling and sketching are much faster to compute than OMP, thus even though the error guarantees are weaker, they are at least tractable and can be integrated into modern DL pipelines, much less can be said for OMP. Authors should try to find some ML settings that computing an optimal sparse weight for AMM is particularly useful, and one could justify the expensive overhead to do so.

**Questions:**

Typos:

* Line 38, Frobenius nomrs -> Frobenius norms.

* Line 466, what is $\odot$? Is this the Hadamard product? If so, the notation established before should be $\circ$.

Questions:

* Is it possible to convert the solution of the convex quadratic program into a feasible solution to the original problem via a rounding scheme?

* What are some ML applications that one could justify to compute an optimal or nearly-optimal sparse AMM?

---

> ### Author Response · Authors · 2025-11-28
>
> We sincerely thank the reviewer for their time and the insightful questions. Below, we tried to address the reviewer points raised on weaknesses and questions.
>
> **Weaknesses and questions**
>
> - (W1)/(Q1) yes we can use solution of problems $Q_k(K)$ into solution of problem $P_k(K)$ with error guarantee better than $u_k^\star$.  The objective function of $Q_k(K)$ is $\phi_k(y)=y^{\top} G^{(k)} y-2(G \mathbf{1}, y)$ and it was basically obtained via
> $$
> {\bf 1}^\top G {\bf 1} + \frac kn \phi_k(y)
>  = {n \choose k }^{-1} \sum_{S\subset \{1,\dots,n\}, |S|=k}({\bf 1} - y_S)^\top G ({\bf 1} - y_S)
> $$
> We denote $y^\star$ be the optimal solution to $Q_k(K)$ or the optimal solution multiple of ${\bf 1}$, or a solution that yields a objective value ${\bf 1}^\top G {\bf 1} + \frac kn \phi_k(y)$ better than a fixed value $u_k$. The above is an average, so there must exist at least one set $S$ of cardinality $k$ such that
> $$
> ({\bf 1} - y^\star_S)^\top G ({\bf 1} - y^\star_S) \leq {\bf 1}^\top G {\bf 1} + \frac kn \phi_k(y^\star)
> $$
> Having $y^\star$ we can try to find such $S$ via rounding/shrinkage or sampling/rejection. An optimal set $S$ can be obtained via binary quadratic optimization by minimizing $$({\bf 1} - b\odot y^\star)^\top G ({\bf 1} - b\odot y^\star)$$ with variable $b\in \\{0,1\\}^n$ constrained to $\sum_{i=1}^n b_i \leq k$. An optimal $b$ is the indicator for $S$. We can also accept a good $S$ (good $b$) by iteratively improving $b$ randomly initialized until the value drop below $u_k^*$. This can be performed via simulated annealing for example. $$~$$
> Another approach in the case $K={\mathbb R}^+$, which we have not discussed consists in adopting the classical sampling methods while enforcing a structure aware approach. This algorithm Structure-Aware Probabilistic Sampling (SAPS) directly uses the solution to a convex relaxation to form a, structure-aware sampling distribution, i.e. use the solution of a relaxed problem as the importance scores for sampling. This embeds the complete interaction structure of $G$ into the sampling process itself.
>
>    - **step 1**: ``Optimal" importance scores: solve a convex quadratic program to find the optimal selection vector $x^\star$: $$ x^\star = {\rm argmin}_{x \in {\mathbb R}^n} ({\mathbf 1} - x)^\top G ({\mathbf 1} - x) ,$$ subject to $$\mathbf{1}^\top x = k,\quad\quad {\rm and} \quad\quad \mathbf{0} \leq x \leq \mathbf{1}.$$ The vector $x^\star = [x_1^\star, \dots, x_n^\star]$ now contains the importance scores and $p = x^\star/k$ is a probability distribution over $\\{1,\dots,n\\}$.
>
>    - **step 2**: Perform Weighted Random Sampling using $p$.
> The deterministic and random approaches discussed previously both requires the complete knowledge of $G$ which is costly. We can also rely on approximation/projection $\widetilde G$ to $G$ for the same algorithms.
>
> - (W2) yes the paper is only concerned with approximation of $AB^\top$ by linear sums $\sum_i x_i a_i b_i^\top$. This need to be more clarified.
>
> - (W3)/(Q2) The paper analysis is general but attempt to separate the settings $K={\mathbb R}^n$, $K={\mathbb R}_+^n$, $K=[0,\xi]^n$ and $K=\\{0,1\\}^n$. This was motivated by ML application, for instance recommender systems where the values of selected weights need to be interpretable. We can elaborate further on this point.
>
> **Typos**
>
> - Yes $\odot$ is the Hadamard product and was denoted $\circ$ before.

---

### Official Review · Reviewer_JgEH · 2025-10-21

**Soundness:** 3
**Presentation:** 3
**Contribution:** 2
**Rating:** 6
**Confidence:** 2

**Summary:**

This paper considers a new approach for approximations to matrix multiplication using convex optimization over an interaction matrix. The approach informally works by relaxing an NP-hard sparse optimization problem to a set of convex quadratic programs parameterized by interaction matrices that interpolates between the full interaction matrix and a diagonal matrix. The authors complement their approach by providing extensive experimental validation of their theory.

**Strengths:**

- Although the paper is dense, the paper is structured well and generally well-written. The paper is also well contextualized with a discussion on related work.
- The theoretical results seem to to be well justified, although I did not check all the proofs in detail.
- The authors additionally provide specific case studies, including those that are do not fit the assumptions of the theory, e.g. binary matrices.

**Weaknesses:**

- It seems like the greedy OMP strategy always outperforms the proposed method.
- The runtime scaling seems potentially high, as we need to run k instances of quadratic programming to compute the solution. Can the authors provide runtime analysis of the method (big O)?
- The guarantees are for just Frobenius norm, which is mentioned as a limitation. It would be nice to see whether this method can also give guarantees for spectral norm (either empirically), or rule this out.
- Data is run on synthetic classes of random matrices. A real-world dataset would be useful.

**Questions:**

- Please give context for Table 1 in the main text, along with additional discussion.
- Can you use a different script for matrices and sets, e.g. sets are script instead of capital like matrices? Currently, the notation is a little confusing.
- I think it would be more helpful to replace the last column in Table 2 with the runtime of each method.

Typos:
- line 38: nomr -> norm
- line 752: equation repeated twice

---

> ### Author Response · Authors · 2025-11-28
>
> We sincerely thank the reviewer for their time and for providing positive feedback on our manuscript. We are encouraged that the reviewer found the paper to be sound and well structured. Below, we tried address the reviewer points raised on weaknesses and questions.
>
> **Weaknesses**
>
> - Yes greedy OMP strategy outperforms the proposed method for the considered case study. We expand here on OMP. For OMP, at the beginning of iteration $k$, we have a selected support $S=\left[i_1, \ldots, i_{k-1}\right]$ and an optimal solution $x^{(k)} \in \mathbb{R}^n$ supported in $S$ that yields the best least square approximation of $A B^{\top}$. The associated residual is given by
> $$
> R_k=A B^{\top}-\sum_{j=1}^{k-1} x_{i,}^{(k)}\left(a_{i,} b_{i, j}^{\top}\right)=\sum_{i=1}^n\left(1-x_i^{(k)}\right) a_i b_i^{\top}
> $$
> Then one has to compute the inner products
> $$
> \Delta_l:=\left\langle R_k, a_l b_l^{\top}\right\rangle_F=\sum_{i=1}^n\left(1-x_i^{(k)}\right)\left\langle a_i, a_l\right\rangle\left\langle b_i, b_l\right\rangle=\sum_{i=1}^n\left(1-x_i^{(k)}\right) G_{i, l}
> $$
> for $l \in[1, \ldots, n] \backslash S$, and pick $l$ with the largest $\left|\Delta_l\right|$ or largest positive $\Delta_l$. We can therefore see that OMP relies completely on the "sequential" knowledge of the matrix $G$ but also the intermediate solutions $x^{(1)}=0, x^{(2)}, \ldots$. We note that for $K=\mathbb{R}$, the part of $x^{(k)}$ supported in $S$ is the optimal solution to $\left|H_S x-b\right|^2$ where $H \in \mathbb{R}^{(m \times p) \times n}$ consists in the $n$ columns which are vectorized version of $a_j b_j^{\top}$ and the right hand side $b$ is the representer to $A B^{\top}$ hence equal to $H \mathbf{1} \in \mathbb{R}^{m \times p}$. This solution has an explicit formula,
> $$
> H_S^{+} b=\left(H_S^{\top} H_S\right)^{-1} H_S^{\top} b .
> $$
> We note that
> $$
> H_S^{\top} H_S=G_{S, S}, \quad H_S^{\top} H=G_{S,:} \quad \Longrightarrow \quad H_S^{+} b=G_{S, S}^{-1} G_{S,:} \mathbf{1}
> $$
> Without a doubt, the structural information that OMP algorithm is using is fully encoded by the matrix G. OMP has the advantage of recovering information sequentially. OMP has exact convergence guarantees or provable quasi-optimal rates depending on properties such as Restricted Isometry Property and low coherence. This corresponds here to well-conditioning of $G_{S, S}$ around the identity for any $S$ of size $k$ or to low $G_{i, j} / \sqrt{G_{i, i} G_{j, j}}$ uniformly over $i \neq j$. The algorithm has also good empirical evidence outside these cases. The upper bounds we have established are valid in any regime, and have relatively good quality as the experiments suggest, for instance comparable to OMP for large $\rho_G$. They provides bounds for any type of convex sets $K$, cases in which adapting or assessing greedy algorithms such as OMP is not obvious.
>
>
> - We can decide to solve only the kth problem. In many settings the minimum $u_k^\star$ is realized with this problem, a simple such setting is $K={\mathbb R}$ and ${\rm diag }[G] = g I_n$. As for runtime, we make the following remark;  for $K=R$ or $K=\mathbb R^+$, we have that
> \begin{align*}
> \beta_s \phi_s( y)
> = \beta_s y^\top G^{(s)}  y-2 \beta_s \langle G \mathbf 1, y\rangle
> =z^\top G  z  - 2  \langle G \mathbf 1, z\rangle + (\beta_s^{-1}-1) z^\top {\rm diag}[G]  z
> \end{align*}
> with $z = \beta_s y$. The program $\beta_1 Q_1(K),\dots, \beta_n Q_{n-1}(K)$ are therefore a family
> of regularized quadratic problems. We can rely on warm starts techniques for sequentially solving these related program.
> The complete knowledge of $G$ is the main challenge, and strategies for solving the same problems but with approximations $\widetilde G $ to $G$ has to be investigated.
>
> - The averaging arguments are straightforward for Frobenius norm leading to analytical formulas. We have not investigated how such technique can be used for spectral norm.
>
> - We experimented with synthetic data mainly to be able to compare on different regime of $\rho_G$ and validate its diagnosis on approximation properties. We of course are willing to run experiments with real-world examples.
>
> **Questions and typos**
>
> We will address them as required

---

### Official Review · Reviewer_ccZV · 2025-10-31

**Soundness:** 3
**Presentation:** 3
**Contribution:** 2
**Rating:** 0
**Confidence:** 4

**Summary:**

This paper studies algorithms for approximating matrix products. The main result is Theorem 4.1 which gives a $O(n^3)$ time algorithm to select at most a size $k$ subset of the outer products of $M = AB^T$ which approximates $M$ up to additive error $\|M\|_F^2$ plus a mysterious term that is not so well defined.

**Strengths:**

The paper draws a nice analogy between relating $AB^T$ to selecting how to weight the outer products in the sum $AB^T = \sum_i a_i b_i^T$, leading to a very intuitive algorithm design.

**Weaknesses:**

Unfortunately I don't see a compelling use case of their particular sampling scheme (but note approximating a matrix product is indeed a fundamental problem).

The reason is that there really are no advantages of this method when we look at the axis of say error guarantees, speed, easiness to compute, and further downstream applications. Let me break this down:

- Error guarantees: The error guarantee of their main theorem already has the $\|M\|_F^2$ term (plus an additional term), where $M$ is the matrix product $AB^T$ we wish to approximate. Note that approximating $AB^T$ by the all zeros matrix would already give this guarantee so this error term is not very meaningful theoretically. The nice thing about existing sampling/sketching based approximate matrix multiplication guarantees is that they *decrease* in error as one samples more rows/columns. This is not true here since there is already a huge $\|M\|_F^2$ term.

- Speed: Again, traditional sampling approaches, eg sampling by row norms, can be instantiated in $O(n^2)$ time for $n \times n$ matrices (we just need to compute row norms!), making them linear time which is extremely desirable in practice. The approach of their main theorem, Theorem 4.1, relies on solving a very complex optimization problem which they say takes $O(n^3)$ time. In that case, one can simply just find the SVD (which is even faster and can be done in matrix multiplication time).

- Easiness to compute: Again, the existing row sampling methods are trivial to implement whereas the proposed method requires solving a very complicated optimization problem.

- Further downstream applications: The advantages of existing sampling/sketching methods is that they can easily be instantiated in other settings such as streaming, which is an advantage the proposed method does not have.

Given these considerations, I really do not see any advantage of the proposed method.

**Questions:**

See above.

---

> ### Author Response · Authors · 2025-11-28
>
> We thank the reviewer for their time, the insightful comments. We thank the reviewer for pointing out these limitations. We tried to address the reviewer points raised on weaknesses and questions.
>
> - Error guarantees: yes, the error guarantee has the $|M|\_F^2$ term plus an additional term. More precisely
> $$v_k^\star(K, G) \le u_k^\star(K, G) := \min_{0 \le s \le k} \left( {{\bf 1}}^\top G {{\bf 1}} + \frac{s}{n} w_s^\star  \right),\quad\quad {{\bf 1}}^\top G {{\bf 1}}=|M|\_F^2.$$ But we have that $w_s^\star \leq0$ as soon as $0 \in K$ and by simply minimizing over the half-direction ${\mathbb R}^+ 1$, we get $$u_k^\star(K, G) \leq \left( 1 - \frac{k}{n} \gamma_k^\star \right) {{\bf 1}}^\top G {{\bf 1}}$$
> which we have viewed as a decay on "total information" quantified by $|M|\_F^2$. The bound $u_k^\star(K, G)$ can be even much smaller. We recall that existing error guarantees although have decay e.g. in 1/k, do involve big constants such as $(\sum_i |a_i| |b_i|)^2$ or $(|A|\_F |B|\_F)^2$.
>
> Let us make the following simple comparison with existing error guarantees. We consider the CR approximation considered in the paper "Drineas, P., Kannan, R., & Mahoney, M. W. (2006). Fast Monte Carlo algorithms for matrices I: Approximating matrix multiplication. SIAM Journal on Computing, 36(1), 132-157". In page 141, the authors establish the bound
> $$\mathbf{E}\left[\left|A B^{\top}-C R\right|\_F^2\right]=\frac{1}{k}\left(\sum_i\left|a_i\right|\left|b_i\right|\right)^2-\frac{1}{k}\left|A B^{\top}\right|\_F^2,$$
> where $k$ is the sparsity level and $\left|a_i\right|$ and $\left|b_i\right|$ are the euclidean norms and $$CR = \frac1k \sum_{i} \frac 1 {p_i} a_i b_i^\top$$ and $p_i=|a_i||b_i|/(\sum_i |a_i||b_i|) $ are optimal selection probabilities. If we assume these euclidean norms are equal to 1, or in general that $\left|a_1\right|\left|b_1\right|, \ldots,\left|a_n\right|\left|b_n\right|$ are all equals, in which case the sampling is uniform, we have equality in Cauchy-Schwartz inequality if applied above, implying
> $$\mathbf{E}\left[\left|A B^{\top}-C R\right|\_F^2\right]=\frac{1}{k} n\left(\sum_i\left|a_i\right|^2\left|b_i\right|^2\right)-\frac{1}{k}\left|A B^{\top}\right|\_F^2=\frac{n}{k} \operatorname{Tr}(G)-\frac{1}{k} \mathbf{1}^{\top} G \mathbf{1}=\frac{n}{k}\left(\rho_G-\frac{1}{n}\right) \mathbf{1}^{\top} G \mathbf{1},
> $$
> In our paper, we provide the bound (formula (7)) given above and also provide the formula (line 907)
> $$\gamma_k^{\star}=\frac{1}{\frac{k}{n}+\left(1-\beta_k\right)\left(\rho_G-\frac{1}{n}\right)}$$
> As a result $$u_k \leq\left(1-\frac{1}{1+\left(1-\beta_k\right) z}\right) \mathbf{1}^{\top} G \mathbf{1},$$
> where $z=\frac{n}{k}\left(\rho_G-\frac{1}{n}\right)$. Since $\left(1-\beta_k\right) \in[0,1]$, we have that
> $$
> \left(1-\frac{1}{1+\left(1-\beta_k\right) z}\right) \leq z \quad \forall z \geq 0
> $$
> In particular showing that
> $$
> u_k \leq \mathbf{E}\left[\left|A B^{\top}-C R\right|_F^2\right] .
> $$
>
> - Speed: yes we agree, computing $G$ exactly and from scratch is costly, especially if $n\gg1$. The matrix $G$ is a central object in our theoretical analysis. In practice, more work need to be done in order to integrate the information encoded by $G$ and still propose tractable strategies, for instance through approximated versions of $G$. We believe that our preliminary analysis offer a new perspective but needs to be further investigated and completed from a fast implementation standpoint.
>
> - Existing sampling methods are straightforward and fast. However, error bounds can be *oblivious* to the interaction between
> the columns of A and B and involve large constants such as $(\sum_i |a_i| |b_i|)^2$ or $(|A|\_F |B|\_F)^2$. Numerical experiments suggests that such methods are ineffective in challenging regimes such as the case $\rho_G \gg 1$, with even exact errors being large.
>
> - Downstream applications: We agree that it is important to keep the intended downstream objective in mind when analyzing such methods and when designing tractable strategies.

---

### Official Review · Reviewer_NJTi · 2025-11-01

**Soundness:** 2
**Presentation:** 2
**Contribution:** 2
**Rating:** 2
**Confidence:** 4

**Summary:**

The paper provides a tight theoretical benchmark for the best achievable error in $k$-term Approximate Matrix Multiplication (AMM), shifting focus from worst-case analysis to instance-specific, structure-aware guarantees.

1.  The approximation error is formalized by defining the Interaction Matrix ($G$). This matrix is used to recast the problem as a Sparsity Constrained Quadratic Program, where the error equation is:
    ($||E||_{F}^{2}=(1-x)^{\top}G(1-x)$)
2.  Since the SCQP is NP-hard, the paper derives a tight, computable upper bound ($u_k^\*$) on the optimal error ($v_k^\*$). This is done by solving a series of tractable convex Quadratic Programs, which are relaxations of the hard SCQP.
3.  The Structure Ratio ($\rho_G$) is introduced to quantify problem difficulty (high $\rho_G$ means more term cancellation). Experiments show that standard AMM bounds fail catastrophically when $\rho_G$ is high, while the new bounds remain accurate.
4.  The bounds are shown to be orders of magnitude tighter than existing theoretical alternatives and are demonstrated to be achievable by greedy algorithms like Orthogonal Matching Pursuit (OMP). A simple, analytical Scaled Identity Bound is also provided as a practical, tight proxy.

**Strengths:**

Below are the key strengths of the paper.

* (s1) The paper contributes by reframing the Approximate Matrix Multiplication error problem. The introduction of the Interaction Matrix ($G$) to explicitly model the cancellation and reinforcement structure within the product sum, and its link to a Sparsity Constrained Quadratic Program, is an innovation.
* (s2) The central theoretical claim, deriving a tight, computable upper bound ($u_k^*$) by using tractable convex relaxations (Auxiliary QPs) of the NP-hard SCQP is logically sound. The work successfully establishes a verifiable theoretical ceiling for AMM performance.
* (s3) The experiments show that prior bounds (from sampling/sketching) are often orders of magnitude looser than the actual achievable error, especially in "hard" instances defined by a high Structure Ratio ($\rho_G$).
* (s4) The inclusion of the Scaled Identity Bound (Proposition 4.3) is a strength. This analytical proxy is shown to maintain high tightness while being significantly faster to compute than the main QP bound.

**Weaknesses:**

Main drawbacks concern computational complexity, limited practical utility, and an unresolved gap between theory and efficient algorithmic implementation.

* (w1) The framework's core requirement is calculating the Interaction Matrix ($G$), which takes $O(n^2(m+p))$ time. This is already too slow for large matrices where AMM is crucial. Furthermore, the main theoretical result, the Auxiliary QP Bound, demands solving $k$ convex Quadratic Programs, adding substantial overhead. This high cost restricts the bounds to offline analysis only, precluding their use in real-time or large-scale optimization.

* (w2) The paper successfully proves that a structure-aware algorithm (like Orthogonal Matching Pursuit, OMP) can nearly achieve the derived tight bounds. However, OMP is slow, requiring iterative least-squares solutions. The paper identifies the fundamental theoretical ceiling but fails to provide a fast, scalable algorithm capable of efficiently reaching this boundary, leaving a disconnect between the theory and practical acceleration.

* (w3)  The bounds strictly apply only to approximations that are a linear combination of $k$ selected outer product terms. This framework does not directly cover or offer optimization guidance for many popular, fast sketching techniques (like Gaussian projection or CountSketch). While the bounds critique the performance of these methods, they do not constructively inform how to improve them within their own distinct mathematical frameworks.

* (w4)  The tightness of the computationally efficient Scaled Identity Bound (Proposition 4.3), which is derived from a highly restrictive analytical constraint ($y=\gamma 1$), undermines the practical necessity of the main theoretical result (Theorem 4.1), which requires solving $k$ complex QPs. This raises a question about whether the added computational expense of the main theorem is justified by the marginal increase in tightness over its simple proxy.

* (w5) Although the small-scale experiments suggest the bounds are tight, the theoretical contribution would be strengthened by a rigorous proof or construction that demonstrates the existence of matrix pairs $(A, B)$ where no $k$-sparse approximation can perform better than the derived bound for a general $\rho_G$. This missing element prevents the bounds from being certified as truly unimprovable.

**Questions:**

Below are the key questions that need to be addressed:

* (q1) Given that calculating the Interaction Matrix ($G$) is $O(n^2(m+p))$, which is often slower than the matrix multiplication being approximated, can the authors propose a fast, randomized approximation of the bound? Specifically, can a bound on $v_k^*$ be derived based on a sampled or projected version of $G$ (e.g., $G \approx G_{sampled}$) that is computed in $O(n \cdot poly(k))$ time? A positive result here would transform the benchmark from a purely theoretical tool into an actionable diagnostic tool.

* (q2) Can the authors theoretically analyze or provide empirical evidence for the existence of a fast, scalable structure-aware algorithm (running in time significantly faster than OMP and exact multiplication) that provably achieves an error within a small constant factor of the derived $u_k^*$ bound? Identifying a path to an efficient, near-optimal algorithm is necessary to bridge the gap between the theoretical ceiling and practical application.

* (q3) The paper provides both the complex Auxiliary QP Bound (Theorem 4.1) and the much simpler analytical Scaled Identity Bound (Proposition 4.3), which is shown to be nearly as tight. Can the authors demonstrate a specific class of matrices (or a particular range of the structure ratio $\rho_G$) where the full minimisation over $s \in \{1, \dots, k\}$ in Theorem 4.1 yields a demonstrably better, non-trivial result than the simpler Scaled Identity Bound?

* (q4) The current bounds apply strictly to linear combinations of $k$ columns selected from the original $n$ columns. How do the theoretical insights and the structure ratio $\rho_G$ relate to or bound the error of popular, non-linear, randomised sketching algorithms (e.g., CountSketch or Subsampled Randomised Hadamard Transform, SRHT)?

* (q5) The paper relies on numerical evidence (the $n=30$ exhaustive search) to claim the bounds are tight. To achieve stronger theoretical completeness, the authors should include a constructive proof that demonstrates the existence of a matrix pair $(A, B)$ for which the optimal error $v_k^\*$ is equal to or arbitrarily close to the derived upper bound $u_k^*$. This adversarial construction is standard practice for proving the theoretical tightness of a new bound. Can a similar result be shown?

---

> ### Author Response · Authors · 2025-11-28
>
> We sincerely thank the reviewer for their time, thorough review and the insightful questions . The comments have helped us explore several new theoretical/practical properties. Below, we tried to address the reviewer points raised on weaknesses and questions.
>
> **Weaknesses**
>
> - (W1) yes, we agree that computing $G$ exactly and from scratch is costly. This motivates relying on proxies which we discuss briefly in question (Q1). Regarding solving $k$ convex problem, we make the following remarks. We can decide to solve only the kth problem. In many settings the minimum $u_k^\star$ is realized with this problem, a simple such setting is $K={\mathbb R}$ and ${\rm diag }[G] = g I_n$. For $K=R$ or $K=\mathbb R^+$, we have that
> \begin{align*}
> \beta_s \phi_s( y)
> = \beta_s y^\top G^{(s)}  y-2 \beta_s \langle G \mathbf 1, y\rangle
> =z^\top G  z  - 2  \langle G \mathbf 1, z\rangle + (\beta_s^{-1}-1) z^\top {\rm diag}[G]  z
> \end{align*}
> with $z = \beta_s y$. The program $\beta_1 Q_1(K),\dots, \beta_n Q_{n-1}(K)$ is therefore a family
> of regularized quadratic problems. We can rely warm starts techniques for solving these related program.
> The above is up to a constant equal to
> $$\|AB^\top -\sum_i z_i a_i b_i^\top \|_F^2 + (\beta_s^{-1}-1) \sum_i |a_i|^2 |b_i|^2 z_i^2.$$ Using OMP or the present approach, we only need exact or estimates to the inner product $$\langle AB^\top, a_i b_i \rangle,\quad\quad\quad i=1,\dots,n$$
>
> -(W2) yes we agree. We note however the following. The objective function of $Q_k(K)$ is $\phi_k(y)=y^{\top} G^{(k)} y-2(G \mathbf{1}, y)$ and it was basically obtained via
> $$
> {\bf 1}^\top G {\bf 1} + \frac kn \phi_k(y)
>  = {n \choose k }^{-1} \sum_{S\subset \{1,\dots,n\}, |S|=k}({\bf 1} - y_S)^\top G ({\bf 1} - y_S)
> $$
> We denote $y^\star$ be the optimal solution to $Q_k(K)$ or the optimal solution multiple of ${\bf 1}$.
> The above is an average, so there must exist at least one set $S$ of cardinality $k$ such that
> $$
> ({\bf 1} - y^\star_S)^\top G ({\bf 1} - y^\star_S) \leq {\bf 1}^\top G {\bf 1} + \frac kn \phi_k(y^\star)
> $$
> Having $y^\star$ we can try to find such $S$ via shrinkage or sampling/rejection. An optimal set $S$ can be obtained via binary quadratic optimization by minimizing $$({\bf 1} - b\odot y^\star)^\top G ({\bf 1} - b\odot y^\star)$$ with variable $b\in \{0,1\}^n$ constrained to $\sum_{i=1}^n b_i \leq k$. An optimal $b$ is the indicator for $S$. We can also find a good $S$ (good $b$) iteratively improving $b$ such as with simulated annealing. See also Question (Q2)
>
> - (W3), we have only observed and discussed empirically the limitations of the various schemes. A complete theoretical comparison
> of the different scheme is not straightforward. We try to combine the best of both world in the answer to Question (Q2).
>
> - (W4), this is discussed in answer to Question (Q3).
>
> - (W5), this is discussed in answer to Question (Q5).

---

> > ### Author Response · Authors · 2025-11-28
> >
> > **Questions**
> >
> >  - Question Q1: Yes, relying on a cheap proxies $\widetilde G$ of $G$ can help to quickly derive upper bounds as described in the paper. Here is a quick break-up of one way to analyze this. Let $\widetilde G$ be a projected version of $G$ and we define
> > $$\Delta  G = G - \widetilde G,\quad\quad\quad\Delta  G^{(s)} = G^{(s)} - \widetilde G^{(s)}.$$
> > Let $y$ and $\widetilde y$ be the optimal solutions to ${Q}_s(K)$ and $\widetilde {Q}_s(K)$ respectively. Then
> > $$y^\top G^{(s)}  y-2\langle G \mathbf 1, y\rangle \leq\widetilde y^\top G^{(s)}  \widetilde y-2\langle G \mathbf 1, \widetilde y\rangle =\widetilde y^\top \Delta  G^{(s)}  \widetilde y-2\langle \Delta  G \mathbf 1, \widetilde y\rangle+\widetilde y^\top \widetilde G^{(s)}  \widetilde y-2\langle \widetilde G \mathbf 1, \widetilde y\rangle$$
> >
> > Similarly
> > $$\widetilde y^{\top} \widetilde G^{(s)}  \widetilde y-2\langle \widetilde G {\mathbf 1}, \widetilde y\rangle\leq y^{\top} \widetilde G^{(s)}   y-2\langle \widetilde G {\mathbf 1},  y\rangle =  - \left( y^{\top} \Delta  G^{(s)}   y-2\langle \Delta  G {\mathbf 1},  y\rangle \right)+ y^{\top}  G^{(s)}   y-2\langle  G {\mathbf 1},  y\rangle$$
> > This implies that optimal values are close in the sense
> > $$|w_s^{\star}(K)-\widetilde w_s^{\star}(K)|\leq {max}\_{{x=y, x={\widetilde y}}}|x^\top \Delta G^{(s)} x-2\langle \Delta G {\mathbf 1},x\rangle |$$
> > We note that $\Delta  G^{(s)} = \beta_s \Delta  G + (1-\beta_s){\rm diag}(\Delta  G)$, implying that in terms of spectral norms
> > $\| \Delta  G^{(s)}\| \leq \beta_s \|\Delta  G\| + (1-\beta_s) (\max_i (\Delta  G)_{ii})$, in turn implying
> > $\| \Delta  G^{(s)}\| \leq \|\Delta  G\|$. In the above, we get
> > $$|w_s^\star(K) - \widetilde w_s^\star(K)| \leq \|\Delta  G\| {max}\_{{x=y, x={\widetilde y}}}( \|x\|^2  + 2\sqrt n \|x\| ) $$
> > The same apply if $y$ and ${\widetilde y}$ are the best solutions multiple of ${\bf 1}$
> >
> > - Question Q2: To address this, we can introduce a new method in the spirit of adopting the classic sampling method to be structure aware: Structure-Aware Probabilistic Sampling (SAPS). This algorithm directly uses the solution to a convex relaxation to form a, structure-aware sampling distribution, i.e. use the solution of a relaxed problem as the importance scores for sampling. This embeds the complete interaction structure of $G$ into the sampling process itself.
> >
> >    **step 1**: Compute ``Optimal" Importance Scores: solve a convex quadratic program to find the optimal fractional selection vector $x^\star$: $$ x^\star = {\rm argmin}_{x \in {\mathbb R}^n} ({\mathbf 1} - x)^\top G ({\mathbf 1} - x) $$ or using a proxy $\tilde{G}$ of $G$, subject to $$\mathbf{1}^\top x = k,\quad\quad {\rm and} \quad\quad \mathbf{0} \leq x \leq \mathbf{1}.$$ The vector $x^\star = [x_1^\star, \dots, x_n^\star]$ now contains the importance scores and $p = x^\star/k$ is a probability distribution over $\\{1,\dots,n\\}$.
> >
> >    **step 2**: Perform Weighted Random Sampling using $p$.

---

> > > ### Author Response · Authors · 2025-11-28
> > >
> > > - Question (Q3): In the case $K={\mathbb R}$, the optimal solution of the quadratic program $Q_s(K)$ is $(G^{(s)})^{-1} G {\bf 1}$ and associated optimal value is $w_s^\star(K) = - {\bf 1}^\top (G (G^{(s)})^{-1} G) {\bf 1}$, hence
> > > $$v_k^\star(K, G) \le u_k^\star(K, G) := \min_{0 \le s \le k} \left( {{\bf 1}}^\top G {{\bf 1}} - \frac{s}{n} {\bf 1}^\top (G (G^{(s)})^{-1} G) {\bf 1} \right).$$
> > > In the case ${\rm diag} (G) = g I_n$, one has
> > > $$G (G^{(s)})^{-1} G = P_s(G),\quad\quad\quad P_s(t) = \frac {y^2}{\beta_s t + (1 - \beta_s)g}$$
> > > We note that if ${\bf 1}$ is an eigenvector of $G$, then the optimal solution $(G^{(s)})^{-1} G {\bf 1}$ is also a multiple of ${\bf 1}$ in which case it coincides with the optimal scaled ${\bf 1}$ solution.
> > > Let $G = \sum_i \lambda_i u_i u_i^\top$ be the eigen-decomposition of $G$, then
> > > $${\bf 1}^\top P_s(G) {\bf 1} =  \sum_i P_s(\lambda_i) (\langle u_i,1 \rangle)^2.$$
> > > For $t$ fixed and any value of $g>0$, the sequence $\frac sn P_s(t)$ is increasing in
> > > $s$, hence $\frac sn {\bf 1}^\top P_s(G) {\bf 1} $ is increasing in $s$. Therefore
> > > $$u_k^\star(K, G) =  {{\bf 1}}^\top G {{\bf 1}} - \frac{k}{n} {\bf 1}^\top P_k(G) {\bf 1} .$$
> > > This is to be compared with scaled identity bound
> > > $$u_k^\star({\mathbb R} {\bf 1}, G) =  \left(1 - \frac{k}{n} \gamma_k^\star \right) {{\bf 1}}^\top G {{\bf 1}}.$$
> > > We are then interested in situations where
> > > $$u_k^\star(K, G)  \ll u_k^\star({\mathbb R} {\bf1}, G) .$$ On the one hand, this is equivalent to
> > > $$({\bf 1}^\top G {\bf 1})^2/({\bf 1}^\top G^{(k)} {\bf 1})=\gamma_k^\star ({\bf 1}^\top G {\bf 1}) \gg{\bf 1}^\top (G (G^{(k)})^{-1} G) {\bf 1}$$ or equivalently
> > > $$({{\bf 1}}^\top G {{\bf 1}})^2 \gg ({\bf 1}^\top G^{(k)} {\bf 1}) \times ({\bf 1}^\top (G (G^{(k)})^{-1} G) {\bf 1}).$$
> > > The previous inequality is insured via Cauchy-Schwartz inequality, in view of ${\bf 1}^\top G {\bf 1} =  (\sqrt{G^{(k)}}^{-1}G {\bf 1})^\top  \sqrt{G^{(k)}} {{\bf 1}}$. If we want the inequality gap to be large, we need not have almost linearity of
> > > $\sqrt{G^{(k)}}^{-1}G {\bf 1}$ and $\sqrt{G^{(k)}} {{\bf 1}}$. In particular ${\bf 1}$ ought to be far from being an eigenvalue of G. We have that
> > > $$G {\bf 1} = (\langle AB^\top, a_1 b_1^\top \rangle,\dots,\langle AB^\top, a_n b_n^\top \rangle).$$
> > > The individual scalar products $\langle AB^\top, a_i b_i^\top \rangle$ need
> > > to be variably distributed. On the other hand, using convexity of $P_k$, one has
> > > \begin{align*}
> > > \frac {{\bf 1}^\top P_k(G) {\bf 1}}{\|{\bf 1}\|^2}
> > > =  \sum_i P_k(\lambda_i) \frac{(\langle u_i,\bf 1 \rangle)^2}{\|{\bf 1}\|^2}
> > > \geq P_k \left(\sum_i \lambda_i  \frac{(\langle u_i,\bf 1 \rangle)^2}{\|{\bf 1}\|^2} \right)
> > > = P_k \left(\frac{{\bf 1}^\top G {\bf 1}}{\|{\bf 1}\|^2} \right)
> > > = \frac 1{\|{\bf 1}\|^2} \frac { \left({\bf 1}^\top G {\bf 1}\right)^2}
> > > {\beta_s  \left({\bf 1}^\top G {\bf 1} \right) + (1 - \beta_s)g \|{\bf 1}\|^2}
> > > = \frac {\gamma_k^* ({\bf 1}^\top G {\bf 1})}{\|{\bf 1}\|^2} .
> > > \end{align*}
> > > This shows that
> > > $u_k^\star(K, G) - u_k^\star({\mathbb R} {\bf1}, G)=-k \Delta_k$ and
> > > $$\Delta_k := \left(\frac {{\bf 1}^\top P_k(G) {\bf 1}}{\|{\bf 1}\|^2}  -\frac {\gamma_k^* ({\bf 1}^\top G {\bf 1})}{\|{\bf 1}\|^2} \right),$$
> > > is a Jensen gap. For $k$ fixed, this gap depends on the curvature of $P_k$, on how
> > > the eigenvalues $\lambda_i$ distributed, and on how the coefficients
> > > $(\langle u_i,{\bf 1} \rangle)^2$ are distributed. For instance, if the
> > > eigenvalues and the coefficients are well spread out, then the gap is large.

---

> ### Author Response · Authors · 2025-11-28
>
> - Question Q4: this is a very interesting and challenging question, that we have not pursued yet. Empirical evidence suggest a strong dependance that needs to be investigated. For example, with an SRHT sketching matrix $S \in \mathbb{R}^{m \times n}$, the expected Frobenius norm error for approximating the product $C = AB^\top$ is bounded by the product of the input norms. Following the analysis in Tropp (2011) and Boutsidis \& Gittens (2013), the bound is:
> $${\mathbf E}\left[ \|AB^\top - (AS^\top)(BS^\top)^\top \|\_F \right] \leq \frac{C}{\sqrt{m}} \|A\|_F \|B\|\_F$$
> where $m$ is the sketch size and $C$ is a constant depending logarithmically on the dimension. This standard bound is *oblivious* to the interaction between the columns of $A$ and $B$. It scales with the "worst-case energy" $\|A\|_F \|B\|_F$.
>
> - Question Q5: The objective function of program $Q_s(K)$ is $\phi_s(y)=y^{\top} G^{(s)} y-2\langle G \mathbf{1}, y \rangle$ and it was derived as follow
> $${\bf 1}^\top G {\bf 1} + \frac sn \phi_s(y)= {n \choose s }^{-1} \sum_{S\subset \{1,\dots,n\}, |S|=s}({\bf 1} - y_S)^\top G ({\bf 1} - y_S)$$
> In particular the  optimal value $u_k^\star(K,G)$ from Theorem 4.1 can be expressed as
> $$u_k^\star(K,G) = \min_{s=0,\dots,k} \left( \min_{y\in K} \left({n \choose s }^{-1} \sum_{S\subset \{1,\dots,n\}, |S|=s}({\bf 1} - y_S)^\top G ({\bf 1} - y_S)\right)\right).$$ This minimum is equal to $v_k^\star$ the minimum of the original program ${\cal P}_k(K,G)$ if for at least one $s=1,\dots,k$ there exist $y$ s.t. all associated $y_S$ with $|S|=s$ minimize ${\cal P}_k(K,G)$. i.e. all values $({\bf 1} - y_S)^\top G ({\bf 1} - y_S)$ are equal and are equal to the $v_k^\star$. We consider the following simplistic cases for which we can prove that $v_k^\star =u_k^\star$ for all $k$.
>
>   - Rank-1 case: we assume that $$AB^\top = \sum_i a_i b_i^\top,\quad\quad\quad  a_i b_i^\top =\alpha_i \times  a_1 b_1^\top,$$ where $\alpha_1,\dots,\alpha_n \in {\mathbb R}$. In this case $AB^\top = \lambda a_1 b_1^\top$ with $\lambda:=\sum_{i=1}^n \alpha_i $, hence $$v_k^\star = 0,\quad\quad\quad\quad k=1,\dots,n.$$ Indeed, the 1-sparse solution $y=(\lambda,0,\dots,0)$ yields exact recovery $\sum_{i=1}^n y_i a_ib_i^\top $ of $AB^\top $.For any $s=1,\dots,n$ fixed, the solution $$y = \frac \lambda s (\alpha_1^{-1},\dots,\alpha_n^{-1})$$ satisfies that for all $S\subset \{1,\dots,n\}$ of cardinality $s$ $$({\bf 1} - y_S)^\top G ({\bf 1} - y_S) = |AB^\top - \sum_{i \in S} y_i a_i b_i^\top|\_F^2=|AB^\top - \frac \lambda s \sum_{i\in S} \frac{a_i b_i^\top}{\alpha_i}|\_F^2=|AB^\top -  \lambda a_1 b_1^\top|\_F^2=0.$$ As explained above, this implies that $$u_k^\star=0,\quad\quad\quad k=1,\dots,n$$
>
>   - Orthonormal case: we assume that $$AB^\top = \sum_{i=1}^n a_i b_i^\top,\quad\quad\quad \langle a_i b_i^\top, a_j b_j^\top \rangle = \delta_{i,j} $$ Let $k=1,\dots,n$ fixed. Any $y \in \{0,1\}^n$ such that $\sum_i y_1=k $ minimize the program $P_k(K,G)$ and yields the minimum $|AB^\top - \sum_{i=1}^n  y_i a_i b_i^\top |\_F^2 = \|\sum_{i: y_i=0} a_i b_i^\top\|^2 = n-k$. Hence $v_k^\star=n-k$. This also shows that $y = {\bf 1}$ and for any $S$ of cardinality $k$ one has$$({\bf 1} - y_S)^\top G ({\bf 1} - y_S) = |AB^\top - \sum_{i \in S}  a_i b_i^\top |\_F^2 = n-k.$$As explained above, we necessarily have $u_k^\star = v_k^\star$.
>
> In the rank-one case $$|AB^\top|\_F^2 = \lambda^2 \|a_1\|^2 \| b_1\|^2,\quad\quad {\rm Tr} (G) = \left(\sum_{i=1}^n \alpha_i^2 \right) \|a_1\|^2 \| b_1\|^2,\quad\quad\rho_G = \frac {\left(\sum_{i=1}^n \alpha_i^2 \right)}{\lambda^2}.$$Although very simplistic, any value in $[1/n,+\infty[$ can be taken by $\rho_G$. The orthogonal case can be used to challenge the approach. We consider the orthogonal case, and assume that $$AB^\top = \sum_{i=1}^n a_i b_i^\top,\quad\quad\quad \langle a_i b_i^\top, a_j b_j^\top \rangle = \alpha_i \delta_{i,j} $$ where $\alpha_i >0$ are not necessarily all equal to $1$. Without loss of generality, we assume that $\alpha_1 \geq \dots \geq \alpha_n$. An optimal solution of program $P_k(K,G)$ is $y=(\alpha_1,\dots,\alpha_k,0,\dots,0)$ and the optimal value is
> $$ |AB^\top - \sum_{i=1}^n y_i a_i b_i^\top|\_F^2=|\sum_{i=k+1}^n a_i b_i^\top|\_F^2=\sum_{i=k+1}^n \alpha_i.$$
> We now compute the value of $u_k^\star$. We have that $G = {\rm diag} (\alpha)$, hence $G^{(s)}= G$ for any $s=1,\dots,n$,
> and $G{\bf 1} =  \alpha$. The optimal solution of program $Q_s(K)$ is ${\bf1}$ and associated values is $-{\bf1} G {\bf1}$. As a result, $$ u_k^\star({\mathbb R}, G) := \min_{0 \le s \le k} \left( {{\bf 1}}^\top G {{\bf 1}} - \frac{s}{n} {\bf 1}^\top G {\bf 1} \right)
> =\left( 1 - \frac{k}{n}  \right) {\bf 1}^\top G {\bf 1}=\left( 1 - \frac{k}{n}  \right) \sum_{i=1}^n \alpha_i.$$ Although very simplistic as well, in such setting a rand-LA approach outperform our approach.

---

### Official Review · Reviewer_cc8b · 2025-11-11

**Soundness:** 3
**Presentation:** 4
**Contribution:** 3
**Rating:** 6
**Confidence:** 3

**Summary:**

This paper studies the fundamental limit of approximating a matrix product $A B^{\top}=\sum_{j=1}^n a_j b_j^{\top}$ using only $k$ of its rank- 1 terms. It shows that the optimal $k$-term approximation error can be formulated as a sparsity-constrained quadratic program in a weight vector $x$, where the error equals $(1-x)^{\top} G(1-x)$ and $G= \left(A^{\top} A\right) \circ\left(B^{\top} B\right)$ ($\circ$ is the Hadamard product) captures interactions and cancellations among terms. Since this SCQP is NP-hard, the authors develop a computable, instance-specific upper bound by introducing a family of auxiliary convex quadratic programs based on an interpolated matrix $G(s)$. This yields a structure-aware upper bound $u_k^{{ }^*}(K, G)$ that closely tracks the true optimum and is significantly tighter than classical AMM bounds. They further derive closed-form analytical bounds that expose the role of the structure ratio $\rho_G= \operatorname{Tr}(G) /\left(1^{\top} G 1\right)$, showing that large $\rho_G$ implies intrinsic difficulty due to strong cancellation. Experiments show that the proposed bounds are much tighter than traditional norm-based guarantees, especially when interactions are strong.

**Strengths:**

1. The paper introduces a novel structural diagnostic for approximate matrix multiplication (AMM) through the definition of the structure ratio $\rho_G=\operatorname{Tr}(G) /\left(1^{\top} G 1\right)$, which quantifies the degree of cancellation among rank-1 components in $A B^{\top}$. This idea provides a new and interpretable way to characterize the intrinsic hardness of sparse or sampled approximations, bridging a gap between geometric structure and achievable approximation accuracy. The formulation of the auxiliary convex quadratic program (QP) and its closed-form Scaled-Id surrogate further represents a conceptually elegant framework that connects theoretical limits, practical computability, and structural properties of the data.

2. The mathematical development is rigorous and well-structured, with clear derivations linking the structural ratio $\rho_G$, the auxiliary QP, and the resulting bounds. The introduction of a hierarchy of relaxations: from the exact NP-hard sparse selection to a convex QP and then to the Scaled-Id analytic form-offers a well-motivated trade-off between theoretical tightness and computational efficiency.

3. The proposed diagnostic provides a new lens to understand why standard sketching and sampling approaches may fail in highly structured or cancellation-dominated settings, emphasizing structural hardness rather than algorithmic deficiency.

**Weaknesses:**

1. A key concept introduced in the paper is the structure ratio $\rho_G=\operatorname{Tr}(G) /\left(1^{\top} G 1\right)$, which provides valuable insight into the intrinsic difficulty of achieving a sparse $k$-term approximation of $A B^{\top}$. While theoretically informative, computing $\rho_G$ exactly requires forming the interaction matrix $G=\left(A^{\top} A\right) \circ\left(B^{\top} B\right)$, which involves constructing two dense Gram matrices and their Hadamard product. This computation incurs $O\left(n^2\right)$ time and memory, making it costly when the number of rank-1 components $n$ is large _i.e._, the regime for which understanding approximation hardness is most relevant. As a result, $\rho_G$ is effective only for theoretical diagnosis (the authors also acknowledged this) when sparse approximation is fundamentally limited by cancellation effects. Its direct evaluation may not scale to large-scale matrix multiplication problems without additional approximation strategies or structure-exploiting techniques.

2. While Figure 1 shows that classical AMM methods can be far from the theoretical optimum, this is demonstrated only for very small matrices ( $n=30$ ), a setting in which sketching-based AMM is not intended to excel and likely amplifies the observed performance gap. In the the larger-scale studies in Figure 2 (e.g., $n=5000$ or $n=8000$), this gap is much smaller as the situation is much more favorable for RandNLA-type algorithms although it focuses predominantly on low- or moderate-cancellation regimes ( $\rho_G \approx 1$). For large (fixed) $\rho_G$, the sparsity vs. accuracy plot for is missing.

3. All empirical evaluations are conducted on synthetic datasets generated under controlled structural assumptions (e.g., induced cancellation, repeated columns, nonlinear transforms). While this is appropriate for isolating the effect of $\rho_G$, the paper does not assess the tightness or practical utility of the proposed bounds on real-world matrices. Moreover, the study does not examine common machine learning settings where AMM is routinely used (e.g., regression, kernel methods, low-rank approximation, neural network layers). As a result, it remains unclear whether the diagnostic insights and bounds carry over to realistic data distributions and practical AMM use cases.

**Questions:**

1. Line 361: Fig. 5 $\rightarrow$ Fig. 1

2. The Scaled-Id bound relies on a single-parameter feasible point $y=\gamma 1$. While analytically convenient, this seems restrictive. Why is this direction theoretically justified as representative of the QP's minimizer, and could alternative feasible vectors yield tighter or more interpretable bounds?

3. Can you give some motivating real-world examples where such structures ($\rho_G\gg 1$) naturally arise?

4. I would like to better understand the practical interpretation of the proposed diagnostic. Suppose we are given data matrices $A$ and $B$, and we can compute $\rho_G$ efficiently. If the computed value of $\rho_G$ is large, does this imply that RandNLA-type sketching or sampling methods are likely to fail, and that one should instead rely on sparse-recovery-style algorithms such as OMP (the fast variants of it) for downstream computations? In other words, is the intended use of $\rho_G$ as a decision criterion for choosing between random projection-based and structure-aware approximation methods?

---

> ### Author Response · Authors · 2025-11-27
>
> We sincerely thank the reviewer for their time and for providing thoughtful and constructive feedback on our manuscript. We are encouraged that the reviewer found the paper to be novel and well structured. The comments have helped us identify several new properties that can make the exposition significantly clearer. Below, we tried address the reviewer points raised on weaknesses and questions.
>
> **Weaknesses:**
>
> - Role of $\rho_G$: we agree that while theoretically informative, computing exactly $rho_G$ is costly. As stated,  $\rho_G$
> is effective for theoretical diagnosis, for instance, for quantifying if a sparse sum $\sum_i w_i a_i b_i^\top$ with the non-zeros $w_i$ are of the same order, can approximate well $AB^\top$. This is the case if $\rho_G$ is small. For $rho_G$ large, it seems ``empirically'' many approximations approaches fail too.  One might be interested in rapidly identifying such regimes.
>
> We recall that
> $$
> {\rm Tr}(G) = \sum_i \|a_i\|^2 \|b_i\|^2,\quad\quad\quad
>  1^{\top} G 1 = \|AB^\top\|_F^2
> $$
> The first is easily computable, the individual norms $\|a_i\|$ and $\|b_i\|$
> are usually computed for Rand-LA methods.
> Moreover, we observe that ${\bf 1}^\top G {\bf 1}$ is upper bounded by
>
> $$( |A|_F |B|_F)^2,\quad\quad\quad(|A|_F |B|_2)^2,\quad\quad\quad(|A|_2 |B|_F)^2,$$
>
> and also by
> $$
> \left( \sum_{i=1}^{min(n,m,p)} \sigma_i(A)^2\sigma_i(B)^2 \right)
> $$
> with the last upper bound being the tightest. This bound can be obtained via von Neumann's trace inequality. In particular, if we have some information on the decay of singular values of $A$ and $B$ and the upper bound turns out to be small compared to ${\rm Tr}(G)$, we will know with certainty that the ratio $\rho_G$ is large.
>
> - Figure1/Figure2: For large (fixed) $\rho_G$, the sparsity vs. accuracy plot for is missing. Yes, the results for such setting are not on the present version, we have the results which we will be happy to add in experiments.
>
> - Application to real world matrices: yes, we agree with the reviewer. We have however discussed and produced preliminary results involving matrices arising in recommender systems, which we have not included. Producing $G$ for some of theses examples is fast, since matrices $A$ and $B$ are sparse, prompting incentive for our approach. The results and interpretation were not included since these examples did not cover all the regimes for $\rho_G$ that we were able to produce synthetically.

---

> > ### Author Response · Authors · 2025-11-27
> >
> > **Questions:**
> >
> > - Question on scaled id bound: In the case ${K=\mathbb R}$, the optimal solution of the quadratic program $Q_s(K)$
> > is $(G^{(s)})^{-1} G {\bf 1}$ and associated optimal value is $w_s^*(K) = - {\bf 1}^\top (G (G^{(s)})^{-1} G) {\bf 1}$, hence
> >
> > $$v_k^\star(K, G) \leq u_k^\star(K, G) := {\rm min}_{0 \leq s \leq k} \left( {\bf 1}^\top G {\bf 1} - \frac{s}{n} {\bf 1}^\top (G (G^{(s)})^{-1} G) {\bf 1} \right).$$
> > In the case ${\rm diag} (G) = g I_n$, one has
> > $$G (G^{(s)})^{-1} G = P_s(G),\quad\quad\quad P_s = \frac {x^2}{\beta_s x + (1 - \beta_s)g}$$
> > and we can show that $\frac sn w_s^\star(K) $ is increasing in $s$ hence
> > $$u_k^\star(K, G) := \left( {{\bf 1}}^\top G {{\bf 1}} - \frac{k}{n} {\bf 1}^\top (G (G^{(k)})^{-1} G) {\bf 1} \right).$$
> >
> > In the case $K={\mathbb R}_+^n$ or $K=[0,\xi]^n$, the best solution multiple of ${\bf 1}$ has the following motivations
> >
> >   - ${\bf 1}$ is the solution of the unconstrained problem.
> >   - it does yield a bound that is better than the sparsest solution ${\bf 0}$.
> >   - factoring out ${\bf 1}^\top G {\bf 1}$ is convenient for explicitly quantifying the improvement ratio.
> >   - the decay rate depends on $\rho_G$ and thus is subject to interpretation.
> >
> > For another fixed $x$, the minimum of the quadratic program
> > ${\cal Q}_s(K)$ restricted to direction ${\mathbb R} x$ is given by
> > $$-\frac {\langle x,G1\rangle^2}{\langle x,G^{(s)}x\rangle}$$
> > One can proposes any $x$ that can yield an analytical formula involving $G$ and is subject to interpretation. We can also
> > restrict to two direction, for example, find the best solution of the form $\alpha {\bf 1} +  \beta v$ for some vector $v$,
> > which amount to solve a quadratic program in dimension $2$. Indeed
> > $$
> > (\alpha {\mathbf 1} + \beta v)^\top G^{(s)}  (\alpha {\mathbf 1} + \beta v) -2\langle G {\mathbf 1}, \alpha {\mathbf 1} + \beta v\rangle
> > = xHx - 2\langle a,x \rangle
> > $$
> > where $x=(\alpha,\beta)$ and
> > $$H = \left(\begin{matrix}{\mathbf 1}^\top  G^{(s)} {\mathbf 1}& {\mathbf 1}^\top  G^{(s)} v \\\\
> > {\mathbf 1}^\top  G^{(s)} v &v^\top  G^{(s)} v\end{matrix}\right)$$
> > $$a = \begin{pmatrix}{\mathbf 1}^\top  G {\mathbf 1}\\\\ {\mathbf 1}^\top  G v \end{pmatrix},$$
> >
> > - Question real-world examples where $\left(\rho_G \gg 1\right)$ naturally arise: from the above analysis, such structure might arise with matrices presenting rapidly decaying singular values. This is prevalent in machine learning, for instance in recommender systems/NLP/networks data.
> >
> > - Question on diagnostic with $\rho_G$ large: yes, the main objective of computing $\rho_G$ or at least having an information about its order helps understand the choice of the right type of algorithms to use. This is what is empirically demonstrated through numerical experiments.

---

### Meta-Review · Area_Chair_Cux2 · 2026-01-06

**Summary:**

The reviews on this paper are very mixed -- with three mildly positive reviews and two negative ones. Ultimately, after looking at the paper myself, although it is a borderline case, I decided not to accept.

The main concern, brought up by most of the reviewers is that the authors' bound gives a sort of approximate lower bound on the performance of any AMM algorithm based on row/column selection. But it does not yield any efficient algorithmic approach that improves on existing bounds for AMM based on random column sampling, or achieves sometihng close to the lower bound that they give. For this reason, while some of the ideas in the paper are interesting, the results feel preliminary and not ready for publication. The paper hints that maybe we can be doing something better for AMM -- but they don't make enough of a step towards finding that sometihng.

Other concerns include a lack of experiments on real world matrices or large input matrices, where AMM is mostly applied. This makes the it less clear if they large gaps between random sampling-based AMM performance and the bound of the paper hold in practical settings.

**Reviewer Concerns:**

The authors did a nice job of addressing many reviewer questions, but ultimately do not address the major concerns mentioned above, which are the ultimate reason for rejection.

**Reviewer Scores:**

I think Reviewer ccZV would have increased their score from a 0, maybe a 2-3, given that the authors clarified a point of confusion where their error bound looked worse than the trivial ||M||_F^2. However, other concerns of this reviewer (no efficient algorithm achieving the improved error bound) remain and are significant.

The authors fail to fully address the concerns of the other negative Reviewer NJTi -- in particular, they don't argue that the improved bounds can be achieved by an efficient algorithm or that the proposed bound is 'tight' and thus characterizes the difficulty of AMM with sampling. I don't think this reviewer would have increased their score.

---

### Decision · Program_Chairs · 2026-01-26

Reject